# Effects of Extracellular Osteoanabolic Agents on the Endogenous Response of Osteoblastic Cells

**DOI:** 10.3390/cells10092383

**Published:** 2021-09-10

**Authors:** Giulia Alloisio, Chiara Ciaccio, Giovanni Francesco Fasciglione, Umberto Tarantino, Stefano Marini, Massimo Coletta, Magda Gioia

**Affiliations:** 1Department of Clinical Sciences and Translational Medicine, University of Rome “Tor Vergata”, 00133 Rome, Italy; giulia.alloisio@alumni.uniroma2.eu (G.A.); chiara.ciaccio@uniroma2.it (C.C.); fascigli@uniroma2.it (G.F.F.); umberto.tarantino@uniroma2.it (U.T.); stefano.marini@uniroma2.it (S.M.); coletta@uniroma2.it (M.C.); 2Department of Orthopedic and Traumatology, “Policlinico Tor Vergata”, 00133 Rome, Italy

**Keywords:** osteoanabolic agents, mechanically induced anabolism, bone remodeling, antioxidant supplements, ossification stimuli, retinoic acid, osteoporosis, resveratrol, exosomes, zinc

## Abstract

The complex multidimensional skeletal organization can adapt its structure in accordance with external contexts, demonstrating excellent self-renewal capacity. Thus, optimal extracellular environmental properties are critical for bone regeneration and inextricably linked to the mechanical and biological states of bone. It is interesting to note that the microstructure of bone depends not only on genetic determinants (which control the bone remodeling loop through autocrine and paracrine signals) but also, more importantly, on the continuous response of cells to external mechanical cues. In particular, bone cells sense mechanical signals such as shear, tensile, loading and vibration, and once activated, they react by regulating bone anabolism. Although several specific surrounding conditions needed for osteoblast cells to specifically augment bone formation have been empirically discovered, most of the underlying biomechanical cellular processes underneath remain largely unknown. Nevertheless, exogenous stimuli of endogenous osteogenesis can be applied to promote the mineral apposition rate, bone formation, bone mass and bone strength, as well as expediting fracture repair and bone regeneration. The following review summarizes the latest studies related to the proliferation and differentiation of osteoblastic cells, enhanced by mechanical forces or supplemental signaling factors (such as trace metals, nutraceuticals, vitamins and exosomes), providing a thorough overview of the exogenous osteogenic agents which can be exploited to modulate and influence the mechanically induced anabolism of bone. Furthermore, this review aims to discuss the emerging role of extracellular stimuli in skeletal metabolism as well as their potential roles and provide new perspectives for the treatment of bone disorders.

## 1. Introduction

Despite its stony appearance, bone is a highly dynamic tissue that undergoes a process of remodeling which is even able to accommodate changing mechanical stress. Precise control of osteogenesis has been a traditional focus of bone cell biology research. The possibility to mechanically enhance bone anabolism is a challenging but very exciting perspective to treat skeletal osteopenic disorders. In skeletal tissue, the extracellular matrix (ECM) occupies most of the volume, assuming a proper three-dimensional form within nanoscale environments which are essential in cell development and maintaining function (through its own reorganization). Cells, responsible for tissue organization, are inherently sensitive to their chemical and physical surroundings. In addition to the intrinsic genetic factors that regulate cell fate, extrinsic signals to cells from the dynamic surroundings are essential in leading cells along proper physiological pathways. Hence, all ongoing processes within specific tissues (e.g., proliferation, differentiation and cell death) are concurrently organized through physical and chemical modulators. This review focuses on the role of extracellular stimuli in skeletal metabolism; the biochemical and cell mechanobiology links are extensively discussed, looking at the effect of soluble osteoanabolic agents such as nutritional supplements, antioxidants and exosomes on endogenous osteoblastic response.

## 2. Mechanoresponsive Skeletal Biology

The present section will describe remodeling processes at the cellular level, focusing on the link between biochemical and biomechanical signaling.

### 2.1. Cells and Extracellular Matrix Organization in Bone

The majority of bone tissue (~70%) is an ion reservoir made up of inorganic calcium salts (hydroxyapatite, Ca_10_(PO_4_)_6_(OH)_2_, crystals) [1]. The second abundant component (~30%) is organic, and it carries negative charges, combining with water to create mechanical properties and the structure of the ECM [2]. The ECM organic component is mostly made up of collagens and proteoglycans, which guarantee elasticity, flexibility and tensile strength [3].

The physiological role of non-fibrous proteins (absorbed into the ECM from the serum) includes strengthening the collagen structure by regulating its mineralization [4]. The principal non-collagenous proteins of the bone matrix are sialoprotein, osteonectin (OCN), osteopontin (OPN) and osteocalcin (OCC) (also known as bone γ-carboxyglutamate protein (BGLAP)), which are rich in negatively charged carboxyl groups (Asp and Glu residues) with a high affinity for calcium ions [5]. In particular, OPN and OCN have been reported to limit crack energy by regulating the size and orientation of hydroxyapatite crystals at the collagen gap regions [6,7]. Likewise, water acting as a plasticizer makes bone tough, compliant and weak and causes the mineral phase to respond with viscoelastic behavior [2].

The remaining minority portion of bone tissue contains specialized cells: osteoblasts, osteoclasts and osteocytes (Table 1, Figure 1). Bone-forming osteoblasts (OBs) and bone-resorbing osteoclasts (OCs) constantly mold the bone architecture (nanosized seed crystals are oriented along lines of mechanical stress), rendering the whole bone tissue a dynamic structure which is light yet resistant to compressive forces. Osteocytes, the most abundant cells of the bone, are traditionally believed to be the master modulators of bone remodeling processes, regulating OC and OB differentiation and thus bone resorption and formation. Interestingly, under physiological conditions, the pressure experienced by osteocytes is postulated to be three orders of magnitude greater than that experienced by osteoblasts, due to amplification by constrained boundary conditions of lacuno-canalicular networks [8].

Finally, an increasing number of growth factors and cytokines such as transforming growth factor β (TGFβ), insulin-like growth factors (IGFs), fibroblast growth factor (FGF) and platelet-derived growth factor (PDGF) have been found to be associated with ECM components [9,10,11] (Figure 1a). The extracellular growth factor signaling can be influenced by the release from the matrix storage and/or by activation of latent forms. An example can be seen in the latent form of TGFβ which can go through a force-dependent activation: when it undergoes integrin-dependent tensile stress [12,13] (see Section 2.3.3). Fatigue-damaged regions of bone send signaling factors to target remodeling regions, and in this way, the remodeling cycle starts, and old or damaged bone can be replaced with new tissue [14].

Hence, skeletal tissue has evolved to elicit bone cells (whose cytoskeletons are strictly anchored to the extracellular matrix network) to discern the wide physical clues on a nanometer scale. These responses may initiate the expression of specific genes or the signaling pathways to adapt their morphology in order to accommodate new functional demands [15]. Loss of physiological ECM mechanical protection may be associated with dynamic alterations that accompany ECM changes as disease progresses (e.g., osteoarthritis (OA)) [16]. Overall, the ECM conveys biochemical and mechanical signals that modulate cell phenotypes not only by acting as a biochemical modulator of direct mechanical forces but also by translating biomechanical cues based on the specific type of surface topography [17].

### 2.2. Bone Remodeling

#### 2.2.1. Mechanical Properties and Structural Modification of Bone Tissue

Bone histology accomplishes bone biological function in two structurally distinct histological types of bone: cortical and trabecular (cancellous), which manages to render it both strong and light. Most of the mature skeleton (~80%) is dense cortical bone (the hard outer layer which is distant from the red marrow) that has a lower rate of turnover and a high torsional resistance. By contrast, trabecular bone (close to the red marrow) makes up the rest of the skeleton. Trabecular bone, which is less dense and more elastic, has a higher turnover rate, and its architecture is organized to optimize load transfer (i.e., high resistance to compression) [18].

The bone modeling process changes the attained peak mass in structure and shape through the independent action of osteoclasts and osteoblast cells. Under normal circumstances, modeling-based bone formation represents a tiny fraction of total bone formation. However, it becomes relevant in mediating adult bone adaptation to permanently changed strain [19,20].

#### 2.2.2. Osteoblast Lineage

The skeletal cell types are illustrated in Figure 1, and Table 1 details their structures, function and regulation.

Osteoblasts share the same common ancestor mesenchymal stem cell (MSC) with fibroblast, myoblast, chondrocyte or adipocyte lineages [21,22]. The major genetic markers for osteoblastogenesis include RUNX2 (runt-related transcription factor 2), ALP (alkaline phosphatase), Col-1, OSX (osterix) and OCN. RUNX2 and OSX are zinc finger transcription factors associated with osteoblast differentiation (see Section 2.1 and Figure 1a) [23,24,25]. While Runx2 expression is key to the progression of osteogenic differentiation, a sustained expression of this protein into later stages of this process has, in fact, a negative effect on the overall differentiation [26]. The differentiation process is also subject to regulation by physical stimuli to ensure the formation of bone that is adequate for the structural and dynamic support of the body [21] (Figure 1a and Table 1).

OBs neatly lay bone matrix proteins which are constantly redistributed along lines of mechanical stress, rendering the skeleton particularly resistant to longitudinal loading forces (about 2000 microstrain (µε)), and enhancing the increase in bone mass, thus changing the bone architecture (see Section 4.2) [27]. Osteoblastic cells comprise a diverse population of cells that include immature osteoblast lineage cells and differentiating and mature bone matrix-producing osteoblasts (Table 1). OBs have been reported either to remain on the bone surface as quiescent bone lining cells or, once entombed within their self-secreted matrix, stop secreting the ECM and differentiate into osteocytes [28] (Figure 1a).

Osteocytes also contribute to ending the remodeling process in response to biomechanical stress and produce sclerostin (SOST) (an antagonist of the anabolic Wnt/catenin signaling in osteoblasts) which inhibits bone formation [29,30] (see Section 3.1.1).

#### 2.2.3. Osteoblast Functions

Mature osteoblasts are one of the major cell types responsible for achieving a balance between bone resorption and the formation of new bone as they produce intercellular signals which modulate the differentiation of distinct types of cells. OBs are specialized bone-forming cells that express parathyroid hormone (PTH) and vitamin D receptors and play several important roles in bone remodeling: (i) expression of osteoclastogenic factors (Figure 1b), (ii) production of bone matrix proteins and (iii) bone mineralization.

Bone formation can be augmented through the increased induction of mesenchymal cells into osteoprogenitors and their subsequent differentiation into osteoid (i.e., bone matrix not yet mineralized), secreting osteoblasts. In addition, activation of quiescent bone lining cells into matrix-producing osteoblasts has been documented as a mechanism for increasing the bone-forming cell population [31].

All bone cell types, except osteoclasts, are extensively interconnected by the cell processes of osteocytes (approximately 15 μm long), forming a complex connected cellular network which is ideally suited for mechanosensation and the integration of local and systemic signals [40] (Figure 2). The transduction of forces into biochemical signals is mediated by dynamic molecular processes [52], whereas the integration of microdamage signals into remodeling signals occurs as the bone remodeling compartment (BRC), specifically isolated, prevents any interference from factors liberated in the marrow space [53].

#### 2.2.4. The Bone Multicellular Unit

The bone remodeling cycle takes place within a temporary anatomical structure named the bone multicellular unit (BMU), which is composed of a local group of cells with a definite lifetime: namely, osteocytes, osteoclasts, osteoblasts and their precursor cells which are supplied by capillary blood [54] (Table 1, Figure 2) (new units are continuously formed as old ones die). The BMU is covered by a canopy of cells (in humans, they are called bone lining cells) which delineate the bone remodeling compartment [55]. The BMU dynamically works as a mechanosensitive module that regulates bone remodeling to prevent and remove fatigue-related microdamage and thus allows adaptation of the bone mass and structure [4]. The number and activity of osteoblasts and osteoclasts are determined by a multitude of factors, such as hormones and cytokines as well as locally produced intercellular messengers under the influence of mechanical stimuli [56] (Figure 2).

#### 2.2.5. The Bone Biochemical Markers

Disruptions of bone homeostasis accompany disorders that include osteoporosis, arthritis and many inheritable skeletal diseases. Imbalances in bone homeostasis result in changes in biochemical marker levels, which faithfully report the grade of organ functions [57]. Along with common blood tests (e.g., blood calcium, PTH and vitamin D) and bone mineral density (BMD) assessment using dual-energy X-ray absorptiometry (DEXA), bone biomarkers (cell type-specific and bone turnover biomarkers (BTM)) are employed to monitor the treatment of disease.

Different classifications of markers have been established to focus on specific skeletal processes. In particular, the BTMs are grouped into two categories: bone formation markers and bone resorption markers, according to the metabolic phase during which they are produced. Furthermore, those markers for the detection of various stages of osteoblastogenesis are classified according to the timing of their appearance as early or late markers. Specifically, the early phase involves the expression of RUNX2 and OSX transcription factors (in association with Wnt signaling). The initiated maturation process [58] allows the subsequent upregulation of downstream growth factors, including vascular endothelial growth factor (VEGF), bone morphogenic proteins (BMP2, BMP4), TGFβ, IGF 1 and IGF2, that play important roles in regulating the expression of late differentiation markers for mature osteoblast phenotypes such as ALP, OCC and OPN [59,60,61]. OCN together with secreted acidic and cysteine-rich proteins is involved in early osteoblast differentiation and osteoclast activity [7].

In addition, the processing of type 1 collagen provides additional markers. Osteoblasts secrete type 1 collagen as an intact molecule containing the N- and C-terminal propeptides, which are subsequently cleaved in the extracellular space. Therefore, N- and C-propeptides of type 1 collagen (P1NP and P1CP) levels reflect the rate of collagen network formation, thus becoming markers of bone formation [62]. Levels of OCC, ALP (bone isoenzyme), P1CP and P1NP in serum are employed in clinical applications for monitoring bone anabolic processes.

Regarding bone resorption markers, enzymes and by-products involved in ECM catabolism have been studied, namely: (i) tartrate-resistant acid phosphatase (TRAP) and cathepsin K that are enzymes upregulated by osteoclasts during the bone remodeling process [63,64]; (ii) carboxy-terminal crosslinked telopeptide of type 1 collagen (CTX-1) and amino-terminal crosslinked telopeptide of type 1 collagen (NTX-1); and (iii) hydroxyproline (HYP) and hydroxylysine (HYL), which are the specific crosslinks formed within structural collagens that are generated from the proteolytic degradation of collagen I and then released into the circulation at a rate proportional to bone resorption activity [57].

### 2.3. Cell Mechanosensing

Mechanical forces direct musculoskeletal cellular activities, altering tissue mass, structure and/or quality. Extrinsically and intrinsically generated mechanical forces load musculoskeletal tissues, and the mechanical signal propagates along the micromechanical environment of resident cells [65]. Mechanical cues sensed by cells are transmitted via the push and pull of specific tethered biomolecules. Indeed, the physical continuity between adherent cells and the extracellular matrix also guarantees that the perceived local physical stimuli can be directly propagated by developing mechanical forces across the cytoskeleton, which can be further transmitted to and between cells regulating intracellular signaling pathways. Notably, compared to soluble ligand-induced signal transduction, mechanotransduction (i.e., the mechanism by which forces are transmitted) can be more than a 1000 times faster along cytoskeletal filaments (within a sub-second to second timescale) [66]. Cell mechanosensing is a bidirectional type of signaling which can be either passive or active. (i) Passive mechanosensing (also known as “outside-in” mechanosensing) is able to perceive extrinsic forces such as tension, compression, shear stress and hydrostatic pressure [67], while (ii) active mechanosensing (known as “inside-out” mechanosensing) can be exemplified by intrinsic forces (when cells reorganize their own cytoskeletons) in cell movement and in cell detection of stiffening and the surface topology of the environment [68,69] (Figure 3a).

#### 2.3.1. Molecular Basis of Mechanotransduction in Mechanosensor Cells

Specifically, the ability of cells to perceive the mechanical signals primarily relies on the presence of transmembrane receptor integrins, which, via focal adhesion (FA) complex, transfer external stimuli through the cytoskeleton first, and ultimately to nuclear lamina. In principle, mechanical deformations of laminin proteins can affect the chromatin structure, thus inducing epigenetic regulation of transcriptional structures that can ultimately translate the physical stimulus into biochemical information (Figure 3a).

Tensile force propagation along subcellular mechanosensory complexes relies on the specific elastic properties of each macromolecular complex. In other words, it can be described by spring constants k_CSK_, k_FA_ and k_ECM_ of the component of the cytoskeleton, focal adhesion complex and extracellular matrix, respectively_._ In general, as with wires, the actual elongation of each domain depends on its intrinsic elastic spring constant. In particular, the strength of the focal adhesion complexes is converted to force-induced conformations within intracellular and/or extracellular mechanosensory molecules depending on the specific elastic properties of the component of the cytoskeleton, focal adhesion complex and extracellular matrix (i.e., their respective spring constants: k_CSK_, k_FA_, k_ECM_) (Figure 3b, right side). Diversity in cell type response arises through the molecular composition, which varies according to isoforms, ratios and geometrical arrangements, this way perturbing the whole mechanical transmission series, thus specifically modulating migration, shape, stiffness and adhesion behavior. In particular, biomechanical sensors constitute a group of specific mechanomolecules that respond to external forces with conformational changes and can be (i) proteins, (ii) specialized subcellular structures such as the primary cilium (which is present in nearly every human cell type [70]) or (iii) blended biomacromolecular structures that interact with cellular proteins, alter the composition of membrane lipids or interact with components of the extracellular matrix or cytoskeleton network [71].

Therefore, cell mechanosensing, whether active or passive, leads to intracellular responses that are transduced through the cells and ultimately result in a tailored context-specific reaction. Moreover, this global mechanism is further complicated by the highly dynamic behavior of cells that can adapt their morphology and cytoskeletal organization in response to mechanical forces. Thus, cellular responses to mechanical forces are mediated by load-bearing subcellular structures (such as the plasma membrane, cell adhesion complexes and the cytoskeleton), which are not static but dynamically interconnected and undergo assembly, disassembly and movement, even when ostensibly stable [52]. In addition, specialized mechanoreceptors can enhance mechanosensation depending on the stimulus frequencies [72].

##### Tethers

The molecular basis of mechanotransduction involves adhesive proteins tethered to the three-dimensional extracellular matrix network. As it is mentioned above, FAs are macromolecular mechanotransducer complexes (containing integrins) that associate the ECM with the cytoskeleton and regulate the transmission of biophysical inputs to sensor cells, triggering the reorganization of actin filaments into contractile stress fibers (Figure 3). Besides protein tethering elements, we also find glycan and lipid tethering components. Transverse and elongated proteoglycan molecules form networks across the pericellular space of the osteocyte, connecting the mineralized matrix to the membrane of the cell and its processes [73]. Various types of lipids, tethering and transmembrane protein molecules, lipid rafts and caveolar formations have been shown to form flexible adjustable signaling facilities within the plasma membrane. All transverse tethering elements seem to be specialized in sensing fluid movement [73].

##### Focal Adhesion Complexes

Due to their critical localization at the cell–ECM interface, transmembrane integrins are mediators of bidirectional signaling, playing a key role in “outside-in” and “inside-out” signal transduction [74] (Figure 3a). Integrins (heterodimeric protein complexes that connect the cell to the pericellular environment) span the plasma membrane and form adhesions with the adjacent pericellular matrix or cells. The transmembrane integrin receptors (with more than 20 members) can recognize the Arg-Gly-Asp (RGD) motif present within ECM proteins (such as fibronectin and vitronectin) tethering the cell cytoskeleton to ECM fibers [75]. When the α β dimer becomes active, the cytoplasmic tail of the β subunit undergoes conformational changes [76]. It is thought that ligands bound to the extracellular domain of integrins may transmit signals by activating intracellular signaling, while the modification of intracellular domains also regulates the binding affinity of extracellular molecules [56,71]. Thus, integrins can connect with other adhesion-associated tethering proteins to form adhesions capable of mediating mechanotransduction signaling cascades. On the one hand, the extracellular domain of integrins allows protein bindings such as fibronectin, collagen and laminin as well as other ECM proteins, while on the other hand, the cytoplasmic tail of the integrins enables interactions with various focal adhesion proteins. However, integrins cannot directly bind to the actin cytoskeleton. The α-actinin proteins crosslink actin filaments to the cytoplasmic tail of the β subunit through proteins such as vinculin and talin [77] (Figure 3b, left side). Among proteins mediating mechanotransduction, there are talin, p130Cas (Crk-associated substrate), paxillin and focal adhesion kinase (FAK) (Figure 3b, left side). Talin and paxillin link to the focal adhesion binding sequence of FAK, and talin associates with the cytoplasmic tail of the β integrin subunit [78] (Figure 3b, left side). Paxillin binds to the cytoplasmic side of the focal adhesion initiating signals since it is a substrate for FAK and src kinase [79]. The FAK tyrosine kinase congregates in areas close to focal adhesions, and its activation induces integrin concentration [80]. Cell membrane receptors rarely act alone; hence, these focal adhesion adapter proteins probably synergistically modulate external signaling and, in cooperation with integrins, integrate diverse signals inducing specific physical stress-mediated gene expression [71]. Overall, when bound, integrins activate a cascade of intracellular signaling pathways, which lead to changes in gene expression and affect most aspects of cell behavior.

#### 2.3.2. Bone Biomechanics

The mechanosensory mechanisms in bone include three systems: (i) the mechanosensor system, which is set up by those cells that are stimulated by external mechanical signals; (ii) the mechanotransduction system, which organizes cells that are interconnected to the extracellular net and mediates the transduction of a physical signal into a biochemical one (see above, Section 2.3); and (iii) the mechanoeffector system, which addresses the transduced signal for the maintenance of bone homeostasis [40].

Lastly, ECM remodeling through the proteolytic degradation (by secreted enzymes such as matrix metalloproteinases and cathepsins) of matrix components has important roles in mechanotransduction as it can stiffen or soften the ECM, modulating the tensile force perceived by the cell and thus the cell response. Overall, throughout life, the inorganic matrix mineralization of bone is frequently remodeled by the coordinated action of bone-resorbing osteoclasts and bone-forming osteoblasts (see above, Section 2.2 and Figure 2), the BMU being the mechanosensitive module which integrates local and systemic signals (see below, Section 3).

#### 2.3.3. Mechanosignal Transductions: Prominent Pathways for the Biomechanics of Bone Cells

The process of converting external mechanical forces into a biochemical response is termed as cellular mechanotransduction. The cellular and molecular mechanisms involved are not yet fully understood, but they are believed to be cell type-specific. Osteocytes have long been proven to be mechanosensor cells [81]. However, in recent years, growing evidence has suggested that also bone lining cells, osteoblasts and MSCs can be mechanosensitive as well.

Following the application of a physical stimulus to osteoblastic cells, cell membranes stretch, and the distinct signals received by specific sensor systems such as integrins, cadherins (i.e., cell-to-cell connectors), stretch-activated ion channels and cilia are integrated (see above, Section 2.3) (Figure 3b, right side). Next, mechanical stimuli trigger specific signal transduction pathways, which largely rely on the mitogen-activated protein kinase (MAPK) signaling cascade [82].

The transient receptor potential (TRP) superfamily of cation channels is able to sense a diverse array of stimuli, such as heat, cold, mechanical loading and osmolarity, playing vital roles in the skeletal extracellular and intracellular Ca^2+^ balance [83]. These channels are generally activated by chemical agonists as well and, in many cases, are believed to serve as “integrators” of physical and chemical stimulants [84]. Notably, the TRPV (vanilloid family of TRP proteins) and piezo channels possess a mechanosensitive nature and are of functional importance in mechanotransduction [85,86,87].

Stretch-activated ion channels and integrin receptors are critical for the transduction of the mechanical signals into biochemical signals inside cells (Figure 3b, right side). During bone remodeling, calcium (a major constituent of the mineral phase) is continuously released into the extracellular environment as a free ion, entering into cells through calcium channels, and RUNX2 acts as a mediator. Phospholipase C (PLC) and inositol-1,4,5-trisphosphate (IP3) signaling are activated and promote the release of Ca^2+^ from intracellular stores.

Besides the calcium signaling cascade IP3, experimental studies have demonstrated the involvement of numerous molecular pathways and mediators in mechanotransduction including GTPases and Wnt/β-catenin signaling [88] (Figure 3b, right side and Table 2) [89]. As a consequence, transcription factor activator protein (AP-1), which potentiates chromatin accessibility, is upregulated; thus, it can upregulate the binding of targeted mechanosensitive growth factors such as bone morphogenic proteins (e.g., BMP2, BMP4), TGFβ and IGF 1 and 2 (which promote growth and differentiation by modulating the mechanistic target of rapamycin (mTOR) pathway) [59,90,91] (Table 1).

As regards osteocytes, prostaglandin E_2_ (PGE_2_), which is secreted after the mechanically induced expression of cyclo-oxygenase-2 (COX-2), is considered an important mediator for load-induced bone formation [92]. Furthermore, mechanically stimulated osteocytes are reported to increase their osteoprotegerin/receptor activator of nuclear factor κB ligand ratio (OPG/RANKL), thus interfering with osteoclastogenesis (see Section 3.2). Additionally, the osteocytic expression of the *Sost* gene (a potent competitive inhibitor of bone formation) can be reduced by mechanical loading [93] through the inhibition of canonical Wnt/β-catenin signaling in OBs. Specifically, SOST interposes itself between Wnt ligands and their receptors such as low-density lipoprotein (LDL) receptor-related protein (LRP) 5/6 and the frizzled (Fz) co-receptor (Figure 3b, right side). This inhibitory effect induces an increase in β-catenin intracytoplasmic levels, which leads to their translocation into the nucleus and stimulation of bone formation [30]. Therefore, SOST acts as a coupling factor between osteocytes and osteoblasts (Figure 2 and Figure 3b, right side, see Section 2.2.2).

In turn, the integrated mechanical signal may impact on a myriad of cell functions: energy metabolism [94], cell motility [95], cell adhesion [96,97], cytoskeleton reorganization [98], cell phenotype [95,99], secretome (e.g., RANK/RANKL balance), nitric oxide (NO), prostaglandin PGE_2_, TGFβ [82,100,101,102,103], proliferation [104], differentiation [91,95] (for further details, see next section).

## 3. Signaling in Bone Differentiation Capacity

New therapeutical investigations in bone regeneration target the modulation of cell differentiative capacity. On the one hand, the advance in mechanobiology has led to the creation of an extracellular environment which can influence differentiation lineages without any need for signaling factors [105]. On the other hand, the Wnt signaling pathway, which plays a strong role in OB differentiation, through its Wnt secretory ligands, could prove relevant to researching new methods of bone treatment.

In osteoblastic cells, the association between Wnt signaling and RANK/RANKL/OPG signaling pathways controls and coordinates osteoclastic bone resorption and osteoblastic bone formation (Table 2, Figure 1a and Figure 2), tuning the differentiation states of bone cells.

### 3.1. Dedifferentiation and Differentiation of Bone Cells Play a Role in Bone Mineralization

#### 3.1.1. Wnt Signaling

Wnt/β-catenin signals are known to play a prominent role in bone resorption. The activation of these signals in OB lineage cells such as OBs and osteocytes induces the expression of OPG and then inhibits OC formation [128]. Wnt signaling is also of interest to regenerative medicine for the design of cell-based therapeutics for controlling the differentiation of MSCs. The Wnt gene family, known for influencing various stages of embryonic development and cell fate determination, induces signals which share molecular regulators with cellular redox-mediated networks [129].

Wnts are secreted glycoproteins (of ~40 kDa weight) involved in the maintenance of stem cells which are crucial for the development and renewal of bone tissue. The so-called canonical Wnt signaling pathway dominates osteoblast differentiation processes (Figure 1a) through the promotion of osteoblastogenesis and osteoprotegerin (OPG) expression [22] (Figure 2).

The binding of Wnt ligands on the cell membrane forms a complex (connecting two specific receptors) (Figure 3b, right side) which, in turn, provokes the stabilization of β-catenin, activating the canonical Wnt signal [22]. On the contrary, in the absence of Wnt, cytoplasmic β-catenin is ubiquitin targeted for degradation, and Wnt gene targeted expression is inhibited.

Non-canonical signaling pathway is a generic term which refers to the Wnt pathways that are not mediated by β-catenin (e.g., Wnt/Ca^2+^ pathway and Wnt/planar cell polarity pathway). Given the large number of Wnt ligands, as well as the increasing array of putative non-canonical pathways and receptors, a crosstalk between the two signaling pathways is not a surprise. The reciprocal inhibition of these pathways has been reported to occur through the competition between canonical and non-canonical Wnt ligands for cell surface binding of Fz. However, the mechanism which may represent a general paradigm underlying the activation of other Wnt signaling pathways yet to be characterized remains unclear [130].

#### 3.1.2. The Effect of Modulation of Wnt Signaling on Bone

The expression levels of competitive Wnt inhibitors (SOST, and DKK (Dickkopf-related protein)−1/2) by osteocytes temporarily control the cycle of bone remodeling [29]. In quiescent bone, osteocytes actively express the Sost gene and DKK inhibitors, preventing further bone formation (see Section 2.2.2 and Section 2.3.3). However, during bone remodeling, the levels of Wnt inhibitors decline, permitting bone formation to occur. During the termination phase (when newly formed osteocytes become entombed within the bone matrix), the osteocytes re-express Wnt inhibitors, which block bone formation [131] (Figure 2 and Figure 3b, right side).

Wnt proteins can suppress apoptosis in osteoblast precursor cells prior to cell differentiation, thus facilitating osteoblast differentiation [129]. Wnt/β-catenin signaling is also important for mechanotransduction, fracture healing and osteoclast maturation [132] (Figure 1 and Table 1 and Table 2). The activation of canonical Wnt signaling leads to improved bone strength, while suppression causes bone loss [133]. Wnt signaling not only stabilizes β-catenin but also activates several members of the GTPases from Rho (Ras homologous protein family), which regulate the final stage of cytoskeletal remodeling.

Finally, Wnt signaling is required for optimal loading-induced bone formation (Table 2). Among the overall 19 Wnt ligands, Wnt1 and Wnt7b are the most load-responsive ones [134]. Therefore, in view of the central role played by the cytoskeleton in mechanosensing, it is possible that the Wnt system may modulate the dynamic cytoskeleton organization [56]. It has also been reported that the physiological response to mechanical loading occurs through the Wnt/β-catenin signaling pathway, enhancing the sensitivity of osteoblasts to further mechanical loading [135]. In particular, vibration-enhanced osteogenic responses in MC3T3-E1 cells have been reported to involve Wnt signaling, which induces a decrease in the RANKL/OPG ratio and levels of sclerostin [136] (see Section 3.2).

#### 3.1.3. Dedifferentiative Capacity of the Osteoblastic Lineage

Interestingly, the potential of osteocytes and osteoblasts to dedifferentiate has been reported under peculiar conditions. In this respect, simulated microgravity treatment of human primary osteoblasts has demonstrated the plasticity (i.e., cellular susceptibility to reprogramming) potential of osteoblasts in the heart, inducing a phenotypic regression accompanied by a loosening of pro-osteogenic specialized functions [95]. It has also been reported that cells embedded in the bone matrix are motile and, once given access to the extra bony milieu, will migrate out of their lacunae. Having left their lacunae, the pre-osteocytes/osteocytes can dedifferentiate, potentially providing an additional source of functional osteoblasts [137].

#### 3.1.4. Regulation of Differentiative Signaling Pathways by Vitamin D

At the endocrine level, several factors can influence the bone remodeling process such as vitamin D (Vit D) that is a member of the class II steroid hormones [138,139].

The biologically active form of the hormone, 1,25-dihydroxyvitamin D3 (1,25[OH]_2_D_3_), exerts its action by binding to a specific nuclear receptor [139], that is, the vitamin D receptor (VDR), whose expression can be regulated by 1α,25(OH)_2_D_3_ itself and by other factors such as PTH, glucocorticoids, TGFβ and the epidermal growth factor [140,141,142].

The VDR acts by binding with vitamin D response elements (VDRE) to modulate gene transcription [143]. 1α,25(OH)_2_D_3_ affects human osteoblast growth and differentiation through both the classic VDR-mediated genomic pathway and membrane receptor-mediated rapid responses [144,145,146,147].

As with other steroid hormones, 1α,25(OH)_2_D_3_ can rapidly stimulate extracellular signal-regulated kinase (ERK)/MAPK activity in osteoblast-like cell lines [148] (Table 1).

The canonical ERK/MAPK pathway can be activated through multiple signals encountered by osteoblasts/osteocytes including those initiated by growth factor receptors such as receptors for insulin, IGF-1, fibroblast growth factor and BMPs [149], ECM/integrin binding and FAK activation [150], related biomechanical stimulation [151] and certain non-genomic actions of estrogens [152].

In the case of osteoblasts, a major ERK substrate is RUNX2, which is known to be regulated by 1α,25(OH)_2_D_3_ [153]. The stimulating action of 1α,25(OH)_2_D_3_ on the ERK/MAPK pathway involves membrane-associated 1α,25(OH)_2_D_3_-dependent signal transduction via the VDR, and protein-disulphide isomerase-associated 3 (Pdia3), a membrane-localized receptor for 1α,25(OH)_2_D_3_ [154,155] which has both genomic and non-genomic effects during osteoblast maturation [156]. It has been reported that Pdia3 mediates the rapid effects of 1α,25(OH)_2_D_3_ on PGE_2_ production, protein kinase C activation and the regulation of genomic changes affecting mineralization in osteoblast-like MC3T3-E1 cells [156]. Protein kinase C (PKC) is involved in homologous VDR upregulation and osteocalcin production in rat osteoblasts [142].

The interaction between 1α,25(OH)_2_D_3_ and BMP2 has been reported to be involved in the regulation of osteoblast marker gene expression and mineralization in MC3T3 osteoblasts in which both VDR and Pdia3 are involved [157]. Moreover, data obtained with wild-type and VDR knockout osteoblasts suggested that 1α,25(OH)_2_D_3_ affects mechanical loading-induced nitric oxide production in a VDR-independent manner [158]. It is unclear whether Pdia3 is involved in this effect.

Besides ERK/MAPK signaling, 1α,25(OH)_2_D_3_ also exerts its action by interacting with the osteoblast differentiation regulatory Wnt signaling cascade via two different routes. 1α,25(OH)_2_D_3_-activated VDR binds, in osteoblasts of various origins, to the promoter of the gene encoding the canonical Wnt signaling co-regulator LRP5, stimulating its expression [159]. Additionally, 1α,25(OH)_2_D_3_ is involved in VDR-mediated downregulation of the Wnt inhibitors, secreted frizzled-related protein 2 (SFRP2) and DKK1, suggesting a possible stimulatory role for 1α,25(OH)_2_D_3_ in Wnt signaling that can suppress adipogenesis while increasing the osteogenesis of MSCs [160]. It has been suggested that the VDR-mediated effects of 1α,25(OH)_2_D_3_ on the ERK/MAPK signaling pathway may modulate the responsiveness of bone to mechanical stimulation [147].

### 3.2. RANK/RANK Ligand Signaling Pathway

The RANK receptor is a tumor necrosis factor receptor specific for the RANK ligand, which is involved in the modulation of osteoclastogenesis (Figure 1b). RANKL/RANK signaling regulates the formation of multinucleated osteoclasts from their precursors as well as their activation and survival in normal bone remodeling and in a variety of pathological conditions. RANKL binding to its receptor, RANK, facilitates the fusion, activation and survival of osteoclastic precursor cells, further driving osteoclast differentiation (Figure 1b), which induces downstream signaling molecules including MAPK, tumor necrosis factor (TNF) receptor (TNFR)-associated factor 6 (TRAF6), nuclear factor kappa-light-chain-enhancer of activated B cells (NF-κB) and c-Fos and, ultimately, the activation of key transcription factors, including the nuclear factor of activated T cell (NFAT)c1, which regulate the expression of OC genes [4]. Although RANKL can be produced by several bone cell types [161], osteocytes are thought to be the masters in sensing variations in load and in stimulating osteoclastogenesis via the production of RANKL [151].

The RANK ligand can bind to both the RANK receptor and OPG, which apparently has no direct signaling capacity. OPG, which lacks a transmembrane domain and acts as a secreted decoy receptor for RANKL, was identified prior to the discovery of RANK/RANKL, and it protects the skeleton from excessive bone resorption by binding competitively to RANKL and preventing it from binding to its receptor, RANK. Therefore, the availability and interaction of RANKL/RANK/OPG determine the efficiency of osteoclastogenesis [162].

The association between Wnt signaling and RANK/RANKL/OPG signaling pathways controls and coordinates osteoclastic bone resorption and osteoblastic bone formation (Figure 1a and Figure 2). In vitro tensile strain applied to OBs induces a decrease in the RANKL concentration and RANKL transcription, whereas it increases OPG mRNA in a magnitude-dependent manner [163].

As reported above (Section 3.1.1), the application of 30–120 Hz vibrations on OB cells has been reported to be beneficial for ossification processes because through Wnt signaling, it is able to decrease the RANKL/OPG ratio and levels of sclerostin [136]. Thus, the RANKL/OPG ratio is a key factor in the regulation of bone resorption, bone mass and skeletal integrity. This ratio varies according to the number of systemic factors (Figure 2), and its signaling pathway is also influenced by vibration treatment [164].

## 4. Mechanical Stimulation in the Recovery of Bone Loss

Mechanical signals such as pressure, gravity, waves and electric and magnetic fields could be employed as anabolic mechanical treatments in bone. Mechanical factors are essential not only for the preservation of bone quality and quantity but also for accelerating bone repair following injuries such as fracture healing or osteointegration. The evolving discipline of mechanomics focuses on physical forces and their impact on the cellular and pericellular molecular mechanisms.

### 4.1. Physical Description of Biomechanics

Forces involved in the cell biology response to an applied stress are complex to define. However, they can be simply described by looking at the measure of cell deformation over time.

#### 4.1.1. The Correspondence between Mechanical Stimulus and Strain

Depending on the deformation mode, the elastic modulus (spring constant: the scaling between the stress and strain of cells) can be described by Young’s modulus, the shear modulus and the compressibility modulus, for linear elongation, shear deformation and isotropic compression, respectively. Thus, a specific strain corresponds to any mechanical stimulus (defined as a change in length relative to the object’s original length) which is directly proportional to the magnitude of the applied stress. The concept of forces in living cells is further complicated as cells show a viscoelastic behavior, which leads to a relaxation of the mechanical stress and to an increase in deformation over time [74] (see Section 4.1.2).

In the mechanical testing of bone tissues and cells, micro- and nano-sensors have been developed to convert forces into mechanical deformation in the ranges of piconewtons. Furthermore, several conditions of the force application technique have been used to probe rheological properties [165]. The response of bone loading at a low magnitude and high frequency in activities such as postural control has been shown to be anabolic to bone. Additionally, high-magnitude and low-frequency impact, such as running, has been recognized to increase bone mass [166].

#### 4.1.2. Concept and Terms Employed to Describe Mechanical Stimuli Applied to Bone

Bone formation, regeneration and degradation processes are stimulated by mechanical strain as a result of the applied mechanical stress. In particular, bone cells are responsive to mechanical forces induced in their precise vicinity through the activity of daily living.

The concepts and terms employed to describe and quantify the types and magnitudes of mechanical properties are reported in Table 3 along with the relationship between forces and deformations.

##### Stress and Strain Characteristics

Bone subjected to external mechanical forces produces strain (structural deformation) which can vary in magnitude and mode depending on the intensity of the applied force. Stress is an indication of the magnitude of force applied to an object, normalized to the area over which the force is applied. It is calculated by force per unit of area (N/m^2^) or pascal (Pa). On the other hand, strain ε (or elongation) is a measure of the deformation resulting from an applied force, and it is expressed as a percentage (see Table 3). Clearly, the relationship between stress and deformation in a system depends on material properties, which correspond to the intrinsic capability of transferring stress/strain.

A solid material, which effectively stores energy during the transfer, is termed elastic, and its stiffness is determined by the modulus (e.g., elastic modulus or shear modulus). On the contrary, fluids are termed viscous as they react by changing their rate and flow in response to an applied force. Cells and tissues have viscoelastic properties because they combine the mechanical properties of solids and fluids.

Stress and strain characteristics vary depending on bone tissue histotype: cortical bone is stiffer than trabecular bone, and thus it can withstand higher stress (~150 MPa) but lower strain (~2%) prior to failure, whereas the porous nature of trabecular bone provides greater elasticity than cortical bone, and thus it withstands lower levels of stress (~50 MPa) but much higher strain (~50%) resistance prior to breakdown [167].

##### Strain Frequency

The strain frequency represents the number of applied cycles per second (1/s or Hz), but it can also be expressed as the number of cycles per minute. The increasing frequency of the strain applied to the bone reduces the minimum effective strain required to stimulate osteogenesis, thus enabling strain-related bone formation to occur at lower relative strain magnitudes (ceasing to intensify beyond 10 Hz due to signal saturation) [167] (Table 3).

##### Strain Rate and Strain Distribution

The strain rate and strain distribution represent the temporal and spatial characteristics of the strain magnitude, respectively. Specifically, the strain rate refers to the temporal change in the strain magnitude within each strain cycle (microstrain per second, µε/s), whereas the strain distribution refers to the spatial change in the strain magnitude across a given volume (Δµε/d) (Table 3).

##### Strain Volume

The total number of daily loading cycles can be quantitatively expressed by the strain volume, which derives from the product between the strain magnitude and the rate frequency for a given loading section. While many combinations of these parameters can return same the strain volume, bone adaptation does not respond linearly. In particular, an increase in the skeletal loading duration does not elicit proportional changes in bone mass formation (Table 3).

### 4.2. Frost’s Mechanostat Theory

In 1892, Wolff postulated, for the first time, that bone remodeling is not only influenced by biochemical factors but also under tight biomechanical control in order to adapt to changing load situations. Almost a hundred years later, the “Mechanostat theory” postulated by Frost extended the theory [168], introducing the dependence of bone formation on the quality and frequency of the mechanical stimulus. Accumulating experimental data had confirmed that, in relation to mechanical usage, different biomechanical loading ranges provoked either bone formation or resorption. The theory proposes that even strains in the 50–100 µε range or less increase BMU activation and defines the strain range of 50–300 µε as optimal stimulation, whereas strains that exceed 3000 µε are believed to provoke microdamage which stimulates BMU-based bone remodeling [169]. An order of magnitude higher (25,000 µε) corresponds to the bone’s fracture strain. As a result, there is a minimum effective strain which is needed to be perceived by bone in order to maintain bone mass, thereby addressing the basic mechanical demand. Furthermore, strain sensed by cells is not inertly transduced uniformly at the cellular level, but a cellular adaptation of the environmental changes occurs. Therefore, the overall strain magnitude varies depending on the frequency and rate of the impulse [167].

### 4.3. Bone Adaptation

The beneficial effects of mechanical stimuli on bone mass can be attributed to the sophisticated capability of bone cells to perceive different types of mechanical stimulations, such as shear, tensile, loading and vibration, and then to translate each specific stimulus into intracellular signals that are finely regulated in time and in space. Thus, bone cells are responsive to mechanical loading, but they can and do adapt over time. Bone adaptation occurs at both macroscopic and microscopic levels, altering the tissue mass and architecture to meet its physiological biomechanical requirements. It has been widely accepted that activity with physiological loading adds bone mass, while disuse or microgravity exposure impedes it [166].

Proper mechanical stimulation on bone cells has been reported to increase osteogenic differentiation and matrix mineralization in vitro [170,171,172], whereas cellular accommodation (mechanical acclimatization) of frequent mechanical loading events creates a prolonged cytoskeletal alteration in bone cells, resulting in longer-term mechanosensitive reductions in response to habitual physical stimuli [167]. Therefore, the adaptation of bone to mechanical loads involves several interacting cell types, signaling molecules and pathways.

In biomechanics, tissue responsiveness to loading or unloading is dependent on both genetic and epigenetic factors [97]. Notably, the osteogenic response tends to become saturated as the period of loading increases without interruption, being more responsive to dynamic rather than static strains [100]. Bone adaptation to mechanical loading is described in a mathematical law as a function of both the strain magnitude and frequency [166].

Since strain loading is dynamic, the strain stimulus can be defined using the Fourier method, as shown in the following equation:$$E=k{\;}_{1}\sum_{t=1}^{n}\in_{1}\;f_{1}$$
where *E* = strain stimulus, *k* = proportionality constant, *ε* = peak-to-peak strain magnitude, and *f* = frequency. Therefore, the bone response to mechanical signals seems to correlate with an increased frequency, meaning that smaller strains induced by lower forces applied more frequently are ample to stimulate bone formation and maintain bone mass.

The interdependence of loading parameters is further complicated by the complex dynamics of timing, where very short refractory periods between cycles of loading enhance bone formation, and where separating the loading into multiple short sessions enhances the bone structure [93,173].

### 4.4. Cell Response to Anabolic Mechanical Treatments

The unique feature of bone, which can heal without scar formation, has fascinated scientist for centuries. These distinctive aspects have led to the exploration of the underlying biochemical and biomechanical mechanisms. The modulation of behavioral mechanics in bone is of interest not only to researchers but also to clinicians and physical therapists. Therefore, a thorough understanding of skeletal metabolism and its anabolic stimulation under physical stimulation is required.

The adverse effects of insufficient mechanical loading in bone healing are critical factors that should be considered when considering orthopedic procedures. Mechanical signals such as pressure, gravity, waves and electric and magnetic fields could be employed as anabolic mechanical treatments in bone. Mechanical factors are essential not only for the preservation of bone quality and quantity but also for accelerating bone repair following injuries such as fractures or osteointegration. In principle, in contrast to systemic pharmacological treatment, the advantages of mechanically delivered strategies are that they are safe (at least at low intensities) and include all aspects of the bone remodeling cycle. Traditionally, internal or external fracture fixation protects skeletal integrity in a non-pharmacological fashion. A further possibility, in addition to fracture fixation, to influence bone healing mechanically is the application of biomechanical stimuli (such as waves administered by LIPUS (low-intensity pulsed ultrasound)) [174].

Important findings regard the frequency dependence of anabolic response. The notion that mechanical signals, in general, and LMHF (low-magnitude high-frequency) vibration, in particular, could serve as an anabolic agent in the clinic and thus help in preventing osteopenia has been proved for osteoporotic patients [175]. The effects of sound LMHF vibrations on conventional culturing systems of osteoblastic cells have been reported to have the ability to promote bone formation and to reduce bone loss. Specifically, it has been shown that these conditions promote osteoblast differentiation through an increase in alkaline phosphatase activity and in vitro matrix mineralization, while three-dimensional cultures of human MSC lines showed increased expression of type I collagen, osteoprotegerin and VEGF [170,171,172]. The mechanism behind the frequency dependence of the osteocyte response remains unclear. However, the specific mechanosensor subcellular component seems to depend on the frequency of the mechanical signal. Considering the viscoelasticity properties of cells, it has been supposed that at a frequency below 10 Hz mechanical deformation may be experienced mostly by cell membrane sensors (as these are less stiff and more deformable than solid intracellular bodies). However, at frequencies above 10 Hz, the movement of the solid nucleus may be prominent in driving the cellular response to vibratory stimuli [164].

## 5. Ossification Coactivators

Another determinant factor, apart from the specific mechanical signals employed as an anabolic agent, is the soluble context. Indeed, hormones and soluble Wnt ligands influence the effect of bone metabolism, acting as coupling agents between biochemical and biomechanical osteoblastic responses. It has been reported that androgen receptor disruption increases the osteogenic response to mechanical loading in male mice [176], while estrogen receptor beta regulates mechanical loading in primary osteoblasts [177]. More importantly, estrogen levels may also influence whether vibrations, loading and mechanical strain generate an anabolic effect on bone cells or not [124,125,127,178,179].

The biochemical coupling of mechanical stimuli into cellular responses represents the most exciting target for modulating mechanotransduction. The identification of molecules involved in mechanotransduction may unveil novel targets for therapeutic intervention that can induce adaptation or have additive effects when combined with mechanotherapies. More interestingly, molecular targeting may sensitize mechanotransductive pathways in such a way that superimposes loading results in synergistic adaptation.

As an example, parathyroid hormone (PTH) therapy employs the PTH derivative to enhance the Wnt/β-catenin pathway in osteopenic patients. In fact, PTH-related ligands are attractive lead compounds for the development of osteoanabolic agents [30]. However, PTH has been reported to play dual roles. On the one hand, continuous hypersecretion of PTH, as it occurs in primary hyperparathyroidism, leads to bone resorption. On the other hand, there is clear evidence that the anabolic actions of PTH have direct effects on osteoblasts and indirect effects mediated by activation of IGF-1 (a pro-differentiating and pro-survival growth factor for OBs) and inhibition of sclerostin (antagonist of Wnt/β-catenin signaling). The activation of cAMP-dependent protein kinase (PK)A accounts for most of the PTH anabolic action, which is triggered by PTH binding to its PTH-related protein receptor [180]. PTH enhances the number and the activation of osteoblasts through four pathways: (i) increasing proliferation, (ii) promoting differentiation, (iii) decreasing apoptosis and (iv) arresting the negative effects of the peroxisome proliferator activator (PPAR)γ receptor on osteoblastogenesis. The synergistic effect of PTH and physical exercise has been observed in preclinical studies. Furthermore, cell- and animal-based studies have indicated an increase in PTH receptor sensitization, which is greater than that induced by summative stimuli [181].

Several extracellular osteoanabolic stimuli are reported to modulate osteoblast differentiation, affecting RUNX2 activity (Figure 4). Hence, osteoanabolic supplements such as vitamins, nutraceuticals, trace elements and endosomes are treated in the present section.

### 5.1. Micronutrients in Bone

Nutrition is critical for optimal bone health and prevention of osteoporosis. Indeed, the role of calcium and vitamin D in improving BMD and reducing fracture risk has been well established [182]. The available data provide clear evidence that the effects of nutrition on bone health are not limited to those resulting from calcium and vitamin D intake. The relationship between vitamins other than vitamin D in bone is complex and seems to be affected by genetic factors, gender, menopausal status, hormonal therapy, smoking and calcium intake. It is possible that nutrient patterns, and not individual foods or vitamins, are important in bone health, which would explain some of the paradoxical results related to individual nutrients.

In addition to macronutrients, the so-called micronutrients present in small quantities or traces in food are required to be constantly part of the diet for bone health. Together with macrominerals such as calcium, phosphorus and magnesium, which have well-known roles in bone health, some microminerals also impact on bone health preservation [183]. They usually support the physiological homeostasis of bone, directly or indirectly influencing its constituent cells. Bone health is positively and negatively influenced by a wide range of trace elements. The physiological activity of trace elements (protective or toxic) in the body might be influenced by several factors, including external factors (nutrition) and internal factors (absorption, metabolism, genetic background, age and gender) [184]. Excessively high concentrations or doses of specific trace elements often lead to an opposite effect to the one desired or a situation similar to when toxic trace elements are present, just as doses that are too low do not lead to any appreciable effect.

Moreover, studies have shown that diets that are high in fruits and vegetables have positive effects on bone mineral status, and that nutrients and vitamins, including A, B complex [185,186], C [187], E [187] and K [188], as well as the homocysteine level [189], are important for the maintenance of bone physiological status.

#### 5.1.1. Vitamin A

##### Effects of Retinoids on Osteoblast Cultures

Vitamin A is a fat-soluble vitamin obtained from the diet either as preformed vitamin A (mainly retinol and retinyl esters) in foods of animal origin or as provitamin A carotenoids in plant-derived foods [190,191]. Inside target cells, retinol is oxidized to *all-trans*-retinoic acid (ATRA) (the bioactive metabolite of vitamin A), which binds cellular retinoic acid-binding proteins (CRABP) and specific nuclear receptors (i.e., retinoic acid receptor/retinoid X receptor (RARs/RXRs)) that, once ligand activated, induce transcription of specific genes [192] crucial for the modulation of differentiation, proliferation, inflammation and apoptosis processes [192,193,194,195]. Several studies have demonstrated the key role of ATRA in the regulation of bone cell function and physiological bone remodeling. However, inconsistent data were reported, and there is controversy around the ATRA benefits, which may depend on the cell source and ATRA bioavailability [192,196,197,198,199].

ATRA is a widely used differentiation drug that can effectively induce the differentiation of osteosarcoma cells; however, the underlying mechanism, in many respects, remains poorly understood [200]. It has been shown that, at micromolar concentrations, ATRA is generally able to stimulate osteoblast differentiation [197,198,201,202,203] and promote in vitro osteogenesis in numerous cell systems, including pre-osteoblasts [204], calvarial osteoblasts [197] and MSCs [201,202]. However, several studies have reported that, at nanomolar concentrations, the effects range from the inhibition of osteoblast differentiation to the downregulation of osteogenic marker genes [204,205,206,207].

With regard to micromolar concentrations, it was found that treatment of the rat pre-osteoblast cell line UMR-201-10B with 1 μM of ATRA resulted in increased ALP activity and mRNA expression of matrix gla protein (MGP) and Col1a1 [204]. Moreover, a study on the effects of pharmacologic (≥1 μM) doses of retinoic acid on primary rat calvarial osteoblasts showed that ATRA reduces cell proliferation and stimulates ALP activity and bone nodule mineralization [197].

In mouse mesenchymal cell line C3H10T1/2, it was found that 1 μM ATRA enhances ALP activity, stimulates mRNA expression of *Alp* and *Runx2* and promotes bone nodule mineralization [198,208]. As it has been observed in animal models, these stimulatory effects seem to be mediated by RARα/RARγ nuclear receptors [208]. In fact, in vivo data indicated that ATRA at high concentrations, or co-treatment of ATRA with BMPs, seems to enhance osteoblast differentiation and function [209].

It was reported that micromolar concentrations of ATRA inhibit mineralization, ALP activity, collagen type I protein and mRNA expression of *Alpl*, *Bgalp* and *Col1a* in primary mouse osteoblasts and MC3T3-E1 cells [210]. Additionally, the mRNA expression and protein level of dentin matrix phosphoprotein 1 (DMP1) is enhanced in MC3T3-E1 cells treated with ATRA [207]. Likewise, in vitro studies on MC-3T3 cell osteogenesis, supplemented or not with 0.5 μM retinoic acid (RA) (the most bioactive form of vitamin A), reported that RA disrupted OB differentiation without affecting ALP activity. However, there was a reduction in Wnt gene expression of *cMyc*, *Lef1*, *Lpr5*, *Lpr6* and *Wnt11* and an increase in Wnt inhibitor expression of *Dkk1* at day 21 and *Dkk2* at days 14 and 21 [211].

Studies of ATRA treatment on human fetal platal mesenchymal cells (hFPMCs) highlighted the importance of the proteolytic remodeling of the extracellular matrix in realizing signaling molecules. Li and coworkers showed that the vitamin A metabolite can dose-dependently inhibit cell proliferation and expression of ECM proteins such as fibronectin, tenascin C and fibrillin2 by modulation of matrix metalloproteinase 2 (MMP-2) and its physiological inhibitor tissue inhibitor of matrix metalloproteinase 2 (TIMP2) through downregulation of TGFβ/Smad (small mother against decapentaplegic) signaling [212].

However, while high-dose ATRA treatment in cultured cells seems to promote osteoblast differentiation, generally, the opposite occurs at low doses.

Lind and coworkers found that ATRA, in the range of 4 to 400 nM, negatively regulates mineralization in both primary human osteoblasts and MC3T3-E1 cell cultures [207].

In particular, ATRA upregulates TNF Superfamily Member (*TNFSF) 11* mRNA (encoding RANKL that supports osteoclastogenesis) while, in parallel, decreasing osteoblast differentiation in MC3T3-E1 cells, by inhibiting cell proliferation and osteogenic gene expression (including *Alp*, *Ocn*, *Runx-2* and *Osx*). A further study in organ-cultured mouse calvarial bones showed that 100 nM ATRA inhibits the expression of a variety of genes associated with both osteoblast differentiation and bone matrix biosynthesis such as *Runx2*, *Sp7*, *Alpl*, Bgalp and *Col1a1* [204].

Moreover, in fetal rat calvarial cells treated with nanomolar concentrations of ATRA, ALP activity, *Bgalp* mRNA and bone nodule mineralization were inhibited [205]. A previous work using the human osteoblastic cell line SV-HFO showed that 100 nM ATRA inhibited osteoblastic differentiation, as demonstrated by the RAR-dependent inhibition of ALP and bone nodule mineralization and increased osteocalcin protein secretion [213]. A decrease in mineralization was also observed under osteogenic conditions and when osteoblastic differentiation was forced with BMP2 in mouse osteoblastic cell line MC3T3-E1 treated with either ATRA, 9-cis retinoic acid or Ro 13-6298 (polyaromatic retinoid, or isotretinoin) at 1, 10 and 100 nM [206]. Notably, the retinoids did not inhibit ALP activity but affected the cell morphology, suggesting that the inhibitory effect on mineralization was not primarily due to the inhibition of bone anabolism.

Next, it is interesting to note that human primary osteoblasts that have been exposed to simulate microgravity display a hampered vitamin A metabolism [95], thus indicating that RA may also play a role in mechanobiology.

Overall, these findings support several in vivo observations that indicate that vitamin A inhibits cortical bone formation without affecting trabecular bone formation, at least in rats treated with supra-physiological levels of vitamin A [206].

##### Effect of Retinoids on Bone Health in Humans

The correlation between retinoid intake, serum retinoid concentration and bone health in humans has been extensively reported with heterogeneous findings, showing positive, negative or negligible effects [192,199].

To protect the adult skeleton, the currently recommended daily allowance (RDA) of vitamin A is 900 μg/day for males, and 700 μg/day for non-pregnant or non-lactating females (NIH Consensus Development Panel JAMA. 2001). Several epidemiological studies which investigated the association between vitamin A and osteoporosis reported a high BMD and low fracture risk in individuals with increased intake of vitamin A and increased serum levels of retinoids [214,215,216]. In contrast, it has been shown that high dietary vitamin A intake in the form of multivitamin supplementation or food fortification is associated with an increased risk of fracture and accelerated age-related bone loss [217,218,219,220,221,222].

Other studies have reported a lack of association between vitamin A intake and fragility fracture [192,223]. Furthermore, in contrast to the above individual observations, a meta-analysis of prospective studies suggested that high retinol intake and blood retinol levels have no effect on total fractures but significantly increase the risk of hip fracture [224].

The effects of retinoids can also be influenced by the vitamin D status. In particular, clinical studies reported that increased vitamin A intake coupled with low vitamin D levels promotes low BMD and skeleton fragility [225,226,227].

In fact, there is emerging evidence on the role of vitamin A as an antagonist of vitamin D in increasing calcium absorption and maintaining homeostatic serum calcium concentrations. Both retinoic acid and 1,25-hydroxyvitamin D share a common nuclear receptor (RXR) following their interaction with RAR and vitamin D receptor (VDR), respectively. Hence, a high vitamin A concentration could reduce vitamin D function [228].

These findings require further validation using healthy animals and various established in vivo osteoporotic and fracture models, also because most of the experimental studies are based upon short-term treatments with high concentrations of vitamin A. There is a need for additional in vivo experiments testing clinically relevant concentrations of vitamin A and retinoids in long-term studies, where effects on bone mass and activities of osteoclasts and osteoblasts are assessed in both cortical and trabecular bone.

#### 5.1.2. Vitamin D

##### In Vitro Effects of 1α,25(OH)_2_D_3_ on Osteoblast Differentiation and Mineralization

The actions of 1α,25(OH)_2_D_3_ on the differentiation of bone MSCs, osteoblasts, osteoblast-like osteosarcoma cells and osteoblast cell lines in tissue culture have been extensively described over the past two decades [147,229,230]. It has been reported that 1α,25(OH)_2_D_3_ is able to regulate bone metabolism and functions by stimulating the production of bone matrix proteins (e.g., collagen, OPN, OCC, matrix Gla protein) and ALP activity, in the course of proliferation and differentiation of human osteoblasts [229]. In fact, osteoblast-related genes such as Secreted Phosphoprotein 1 (SPP1) [231], bone sialoprotein (BSP) II integrin binding sialoprotein (IBSP) and RANKL [185] were all shown to contain VDRE binding motifs that could be regulated by 1α,25(OH)_2_D_3_ in isolated osteoblasts. Moreover, 1α,25(OH)_2_D_3_ is an important regulator of RUNX2, with which it cooperates in inducing the expression of OCC [153,232,233] (Figure 5).

Besides the stimulation of mineralization, 1α,25(OH)_2_D_3_ also induces activin A, a strong inhibitor of mineralization. Thus, mineralization induction by 1α,25(OH)_2_D_3_ may actually be controlled via interplay with activin A and OCC, preventing excessive and pathological mineralization [234]. In vitro 1α,25(OH)_2_D_3_ supplementation of aged OBs is also able to offset the reduction in OCC and AP mRNA levels [235]. Interestingly, one study reported that the effects of 1α,25(OH)_2_D_3_ on human OBs are not restricted to classical VDR-mediated transcriptional responses but also involve microRNA (miRNA)-directed posttranscriptional mechanisms, resulting in the regulation of Col4a1 and BMP2K [236].

Nevertheless, it is now well established that 1α,25(OH)_2_D_3_ can both positively and negatively regulate the expression of osteoblast phenotypic markers as a function of the proliferative and differentiated states of osteoblasts and the duration and concentration of exposure [147,234]. An early study by Owen and coworkers showed that acute 1α,25(OH)2D3 treatment inhibits proliferation but strongly stimulates matrix Gla protein and *Spp1* expression in early cultures of rat calvarial osteoblasts, while the same treatment stimulates osteocalcin and mineralization in differentiated cells [237,238].

Important information on the transcriptional response to 1α,25(OH)_2_D_3_ in osteoblasts comes from genome-wide expression profiling studies. Using this analysis, 1,25(OH)2D3 treatment of murine MC3T3-E1 cells has been shown to downregulate DNA replication genes [239], whereas this same gene set was not affected in human OBs [234].

In this regard, Woeckel et al. demonstrated that, apart from indirect effects via intestinal calcium uptake, 1α,25(OH)_2_D_3_ directly accelerates osteoblast-mediated mineralization via the increased production of ALP-positive matrix vesicles in the period prior to mineralization, which leads to an earlier onset and higher rate of mineralization. These effects are independent of changes in the extracellular matrix protein composition [234]. Overall, these studies emphasize the importance of considering the differentiation stage when examining responses of mesenchymal stem/stromal cells and osteoblasts to 1α,25(OH)_2_D_3._

In a more recent study, it was found that, 24 h after treatment of human OBs with 1α,25(OH)_2_D_3_, most genes were upregulated, indicating predominant transcriptional activation by this hormone [240]. Pathway analyses identified various functional gene categories related to bone metabolism and skeletal development [241]. Notably, in human and mouse osteoblasts, 1,25(OH)_2_D_3_ induces the expression of the odd-skipped-related genes *Osr1* and *Osr2*, both known to be expressed in developing limbs [241].

It is important to emphasize that the 1α,25(OH)_2_D_3_ effects on osteoblast differentiation and mineralization may be dissimilar according to the animal species considered. In particular, a discrepant responsiveness has been shown between human/rat osteoblasts and murine osteoblasts, with the effects, overall, being stimulatory in human and rat osteoblasts and inhibitory in murine osteoblasts [229,242,243].

Similarly, in human osteoblasts, 1α,25(OH)_2_D_3_ has been shown to increase *Runx2* expression [244,245], whereas in murine osteoblasts, 1α,25(OH)_2_D_3_ suppresses the RUNX2 promoter and inhibits *Runx2* expression [153]. Additional studies have reported diametrically opposing responses in the vitamin D regulation of the mouse vs. the human and rat osteocalcin genes [246]. In contrast to human and rat osteoblasts in which 1α,25(OH)_2_D_3_ stimulates *Bgalp* expression, 1α,25(OH)_2_D_3_ inhibits BGLAP expression in murine osteoblasts [247,248]. A full explanation for these discrepancies is lacking. The extracellular milieu as well as the intracellular milieu of the cell may contribute to the differences in the 1α,25(OH)_2_D_3_ effects observed in human and murine osteoblasts. In this respect, it is well established that 1α,25(OH)_2_D_3_ and VDR regulate gene transcription in osteoblasts via interaction with a multitude of other transcription factors and DNA- and histone-modifying proteins [249,250]. It is therefore important to consider the effects of 1α,25(OH)_2_D_3_ on osteoblasts in the context of interaction with other hormones (for example, PTH or cortisol) [142,251], growth factors such TGFβ, IGF-1, BMPs, interferons, hepatocyte growth factor (HGF) and epidermal growth factor (EGF) [157,234,252,253] and other signaling molecules such as the peroxisome proliferator-activated receptor ligand rosiglitazone and Wnt signaling [234].

Alternatively, 1α,25(OH)_2_D_3_ may modulate the activity of other hormones, factors and signaling cascades. 1α,25(OH)_2_D_3_ enhanced, for example, the 17β-estradiol effect in female but not male human osteoblasts, as assessed by an increased creatine kinase response [254].

These data together with the discussed differences in in vitro mineralization and osteocalcin expression in osteoblasts warrant a careful interpretation of the data when considering the human situation [255,256].

##### Vitamin D Status and Bone Health

It is widely agreed that vitamin D is essential for bone health; inadequate vitamin D intakes over long periods of time can lead to bone demineralization, resulting not only in the classical deficiency diseases of rickets and osteomalacia but also in increased bone metabolism and enhanced fracture risk [257,258]. Nevertheless, the degree to which vitamin D directly affects bone vs. its indirect actions via 1,25(OH)_2_D stimulation of intestinal calcium and phosphorus absorption remains a matter of debate, although both are clearly involved [259,260].

In vivo models have shown direct effects of 1,25(OH)_2_D on various bone cells, which suggests a direct effect [261]. On the other hand, vitamin D deficiency in animals (and humans) that lack a functional VDR or cytochrome P450 family 27 subfamily B member 1 (CYP27B1) can be successfully treated by increasing the calcium and phosphate content of the diet [262,263].

In fact, numerous studies have demonstrated the close interaction between calcium and vitamin D with respect to their compensatory/synergistic actions [264].

In this regard, the vitamin D metabolites have a multitude of effects on systemic calcium homeostatic mechanisms, which themselves impact on bone. A lack of vitamin D results in hypocalcemia and hypophosphatemia, which is sufficient to cause rickets [258,265]. Vitamin D metabolites can also alter the responsiveness of bone to growth hormone [266] and the expression and/or secretion of a large number of skeletally derived factors including IGF [266], TGFβ [267], VEGF [268], interleukin (IL) 6 [269] and 4 [269] and endothelin receptors [270], all of which can exert their own effects on bone as well as modulating the actions of the vitamin D metabolites on bone.

Nevertheless, a full understanding of the impact of vitamin D metabolites on bone is complicated by species differences and differences in responsiveness of bone cells according to their states of differentiation (see Section 5.1.2). Moreover, a number of additional parameters such as diet (that is, composition and concentrations of minerals), age, sex, timing of treatment, duration of treatment and dosages should be taken into account when comparing in vivo studies, although they are often missing or not reported in sufficient detail [264].

Given the relationship between vitamin D and bone health, an optimal vitamin D status is essential for the minimization of fracture risk and prevention of bone-related disease. The dietary intake of vitamin D required to prevent vitamin D deficiency and ensure an optimal vitamin D status will vary depending on sun exposure preferences. Although the optimal level of vitamin D to maintain bone health remains under debate, the majority of trials and meta-analyses indicate that a dose of vitamin D of 800 IU per day is required to achieve the 30 ng/mL (75 nM) level of 25[OH]D that is recognized as safe and effective [271].

Supplemental calcium may enhance the beneficial actions of vitamin D on bone [272]. Reports of toxicity have arisen from excessive dietary intakes of the vitamin, with all such cases reporting serum 25[OH]D concentrations of >200 nmol/L [263].

Evidence indicates that supplementation with vitamin D in those most at risk of impaired bone health has a beneficial effect on fracture prevention [273]. These benefits are a combination of increased intestinal calcium absorption [274], increased BMD [275] and reduced risks of falls [275].

Emerging evidence clearly suggests vitamin D also has the potential to modulate the effect of pro-inflammatory cytokines on bone metabolism [243,265].

#### 5.1.3. Vitamin K Status and Bone Health

Vitamin K is the collective term for a family of fat-soluble compounds that share a common 2-methyl-1,4-naphthoquinone ring but which differ in the side chain at the 3-position. The three main forms are vitamin K1 or phylloquinone (PK), vitamin K2 or menaquinone (MKn) and vitamin K3 or menadione. MK-4 is the predominant form of vitamin K2 in the human body [276,277,278,279].

To date, there is insufficient evidence to determine the estimated average requirement for vitamin K [280], and consequently, recommendations are inconsistent. The Institute of Medicine has proposed an adequate dietary intake for men and woman of 120 and 90 μg/day, respectively (Institute of Medicine (US) Panel on Nutrients, National Academy Press, 2001). Vitamin K is mainly known as an essential factor in blood coagulation. In addition, it has also been found to have many other functions, and emerging evidence indicated that vitamin K may have a protective role against age-related bone loss [281].

It can modulate bone metabolism through several mechanisms. Firstly, and as the most well-known mechanism, vitamin K acts in the endoplasmic reticulum as a coenzyme for the gamma-glutamyl carboxylase (GGCX) enzyme, which carboxylates glutamic acid (Glu) residues in vitamin K-dependent proteins (VKDPs), transforming them into gamma-carboxyglutamic acid (Gla) [282]. There are several relevant VKDPs in bone, including matrix G1a protein (MGP), periostin, Gas 6, protein S and OCC (or bone Gla protein) [283,284,285]. Osteocalcin has three Glu residues, and its binding capacity depends on its degree of carboxylation. However, full carboxylation of Glu residues is not the normal state of osteocalcin in human bone tissue. Several studies have reported that low serum K1 concentrations, high levels of undercarboxylated osteocalcin (ucOCC) and low dietary intake of both K1 and K2 are associated with a higher risk of fracture and lower BMD [286,287,288]. Interestingly, comparing pharmacodynamic and pharmacokinetic properties of K1 and MK-7 supplements, it has been shown that MK-7 induces a more complete carboxylation of OCC, suggesting higher effectiveness [289,290].

Another vitamin K-dependent protein is matrix G1a protein, which is secreted by chondrocytes and vascular smooth cells and exerts its role as an inhibitor of angiogenesis and ectopic tissue calcification [291]. G1a-rich protein and periostin regulate extracellular matrix mineralization, and protein S, although mainly known for its role in coagulation, also plays a role in bone turnover, although its pathways are unclear [188].

In addition to gamma-carboxylation, vitamin K plays an important role in bone via other mechanisms. It can regulate the genetic transcription of osteoblastic markers, can suppress bone resorption and can regulate the formation of osteoclasts [277].

Vitamin K activates the nuclear steroid and xenobiotic receptor (SXR), also known as Pregnane X Receptor (PXR), a murine homolog, inducing the expression of its target genes in osteoblastic cell lines. In particular, SXR/PXR forms heterodimers with the 9-cis-retinoid acid receptor (RXR), and this latter complex binds to SXR-responsive elements within target genes [292].

The genes induced by vitamin K in an SXR-dependent manner include tsukushi (Tsk), matrilin-2 (Matn2) and cluster of differentiation protein CD14 [292]. Tsk encodes a protein that has a collagen-accumulating effect, and Matn2 is a protein comprising an extracellular matrix such as collagen, whereas CD14 regulates osteoblastogenesis and osteoclastogenesis by inducing the differentiation of B cells [293]. Thus, the activation of SXR/PXR in bone tissue promotes bone formation and suppresses bone resorption, indicating that SXR/PXR may be a key regulator of bone homeostasis [277].

In addition, in vitro and animal studies have shown that MKns are able to inhibit osteoclastic bone resorption, by suppression of RANKL expression [294]. In particular, MK-4 may be involved in inflammation [295], oxidative stress and apoptosis, all of which can inhibit bone reabsorption. Additionally, an in vitro study showed that MK-7 suppressed osteoblast differentiation and stimulated the mRNA production of osteocalcin, osteoprotegerin and RANK-L [296].

There is a consistent line of evidence that vitamins K and D work synergistically on bone density and development [297]. Notably, it has been shown that vitamin K2 enhances vitamin D3-induced mineralization, possibly through the accumulation of osteocalcin in the extracellular matrix of human osteoblasts. It also has been seen to increase osteocalcin gene expression [298].

It has also been reported that supplementation with vitamins MK-7 and D3 and a combination of both is able to modulate the expression of genes involved in both mineralization and angiogenesis, and that vitamin MK-7 enhances the vitamin D3 effects on human MSCs [297]. Further in vivo studies should be conducted to assess how these molecular effects translate into accelerated bone healing [297].

Several epidemiologic studies have investigated the association of vitamin K status and various markers of bone health, including clinical endpoints such as BMD and the fracture rate. These studies revealed that vitamin K deficiency is related to osteoporosis, pathological fractures and vascular calcifications, suggesting a beneficial effect of vitamin K on bone health [286,299,300,301,302].

Notably, a key finding was that vitamin K supplementation has a positive effect on the skeleton of postmenopausal women with a reduced incidence of fracture, mediated by mechanisms other than BMD increase [286,299]. Whether higher vitamin K intakes are associated with higher BMD values, however, remains a controversial matter [188,277,283]. Overall, it must be stressed that evidence from clinical trials is still scarce and limited, and thus controversy remains over the use of vitamin K1 and K2 supplements, which makes it difficult to arrive at solid conclusions. High-quality clinical trials are needed to confirm the current results and to make a specific, practice-changing recommendation.

#### 5.1.4. Zinc as an Emergent Ossification Stimulus

Elements can be classified as essential or partially essential depending on the level of involvement in bone function, while others have been identified as toxic. Zinc and copper suppress bone resorption, promote bone formation and increase bone density and quality. Moreover, iron, boron and fluoride also have bone-protective effects. In contrast, cadmium, chromium and cobalt have toxic effects, even in small concentrations (Table 4). However, even bone-protective elements (zinc, fluoride, magnesium, iron) can also have undesirable effects on bone health in case of excessive intake. The biological relevance of these nutrients arises from numerous studies that associate nutritional deficiency of essential elements, as well as high exposure to toxic elements, with severe skeletal disorders and the subsequent occurrence of pathological conditions involved in bone remodeling and reduced regenerative capacity [184].

For this reason, a better understanding of the cellular mechanisms underlying their effect in the bone context would make it easier to establish their specific influence on bone anabolism. In the present section, zinc will be analyzed as an important trace element in bone physio-pathological conditions, regarding both the bulky in vitro and in vivo evidence where it was employed as an exogenous bone inducer. Therefore, it is considered as a promising trace element in promoting the production of bone mass [326].

Among trace minerals, zinc is the second most abundant transition metal in organisms, after iron, which is essential for various cellular processes, playing catalysis, regulation and structural roles [327]. It is usually obtained through the diet, and following intestinal absorption and plasma transport by albumin and transferrin, it is distributed in different percentages throughout the body [328]. To protect the adult skeleton, the currently recommended daily allowance (RDA) of zinc is 11 mg/day for males and 8 mg/day for females [183]. Of the amount of zinc intake from the diet, skeletal muscle is the main reservoir (60%) followed by bone (∼30%), the liver, the skin (∼5%) and other tissues (2–3%) [329]. However, bone zinc levels are considered the best indicator of total body zinc levels, in plasma and in other organ storages [330].

In the skeleton, zinc localization in the mineral component raises the apatite crystal content [331], while within cells, zinc homeostasis is regulated by the Zrt- and Irt-like protein (ZIP) family and zinc transporter (ZnT) (see Section 5.1.4). The ZIP and ZnT families act as importers and exporters of zinc, respectively [332]. Although the total intracellular zinc concentration is in the range of 100 to 500 Mm, 90% of total zinc is tightly bound to proteins, where it acts as a cofactor for approximately 300 enzymes and hormones [333], while free zinc ion (within the range of 10–100 pM) acts as a second messenger for numerous signaling pathways [334].

Overall, zinc is dynamically stored between the mineral and cellular components of the bone, and therefore it is released from the reservoir during the breakdown of the skeleton, whereas it is incorporated during bone formation. Consequently, it is not surprising that zinc is primarily involved in bone growth, mineralization and regeneration (Figure 6), affecting mainly, but not exclusively, osteoblast biology [326].

As previously reported, bone mass apposition processes require the expression of several genes, including early and late differentiation markers typical of bone [36] (see Section 2.2.5).

The multitude of signaling pathways involved in this process is tightly regulated at the transcriptional level by zinc finger transcription factors (ZF-TFs), which require structural zinc to maintain their integrity and DNA-binding functionality [335] (Figure 7). The two major families of zinc finger transcription factors are Kruppel-like factors (KLFs) and specificity proteins (Sps), both involved in the regulation of gene expression in osteoblasts through interactions with multiple transcription factors [336].

It is therefore intuitive to think that the presence of zinc enables the transcription regulation of differentiation genes, while its depletion is related to a deficit in this process. Thus, several studies both in vitro and in vivo highlighted the anabolic role of zinc in regulating bone turnover, suggesting it as an excellent osteogenic element.

##### Cell importers and Cellular Transporters of Zinc

Since zinc ions cannot freely pass through lipid bilayers, zinc’s influx and efflux are mediated by specific membrane transporters, which regulate its cellular homeostasis [337]. At the genetic level, the Slc39a family of importers encodes ZIPs, and the Slc30a family of exporters encodes ZnTs [332].

Structurally, ZIPs are homo- or heterodimers with eight transmembrane domains and their N-terminal and C-terminal regions located extracellularly, whereas ZnT transporters have six transmembrane domains, and the N-terminal and C-terminal regions are cytoplasmic (Figure 7). In both cases, the presence of a histidine-rich loop mediates their binding to zinc and their subsequent transmembrane transport [338].

Thus far, very few studies on Zn transporters have been reported. Nonetheless, some proteins (ZIP1, ZIP8, ZIP13, ZIP14, ZnT5 and ZnT7) have been identified to play a key role in bone homeostasis. Not surprisingly, the abnormal function of these ZIP and ZnT zinc transporters causes dysregulation of zinc homeostasis, contributing to human bone diseases [339].

The ZIP1 importer has a ubiquitous membrane location in osteoblasts, and during the differentiation process of MSCs into osteoblast-like cells, its protein expression is increased, and consequently, the cytoplasmic zinc influx is increased. This influx allows an upregulation of the key osteogenic regulators RUNX2 and OSX, which, in turn, modulate the transcriptional expression of ZIP1 by directly binding to the responsive elements in the promoter [340]. Indeed, as highlighted by a study carried out on osteoblastic cell line MC3T3-E1, high zinc exposure increases the expression of ZIP1, which allows a considerable influx of zinc [242]. Accordingly, studies showed that induction of ZIP1 gene overexpression in MSCs induces increased mineralization as well as increased expression of the differentiation marker APL and several differentiation genes such as OPN, Cbfa1/RUNX2 and BSP [341].

ZIP1 is also ubiquitous in the osteoclast precursor membrane. Studies that focused on its overexpression showed the blocking of the differentiation process through inhibition of the NF-κB pathway, suggesting that ZIP1 negatively regulates osteoclast function [342].

ZIP14 is localized at the plasma membrane of cells of numerous tissues. Despite belonging to the zinc transporter family, it is not selective for this ion but is able to bind and transport Fe^2+^ and Mn^2+^ [343]. Given its heterogeneous expression on multiple cell types, studies on Zip14 knockout mouse models have shown multiple alterations in different organs, including the liver, adipose tissue, brain, pancreas and bone. In bone, under physiological conditions, ZIP14 is highly expressed in the proliferative zone of the growth plate, in chondrocytes, which are important for bone elongation [344]. Accordingly, in vivo studies showed that Zip14 KO mice exhibited abnormal chondrogenesis and endochondral ossification, osteopenia in both trabecular and cortical bones, dwarfism and scoliosis [345], due to a decreased zinc influx that consequently leads to the inhibition of the CREB signaling pathway, which is involved in osteoblastic differentiation and in the induction of endochondral ossification [346]. Cranial internal hyperostosis (HCI) is a rare bone disorder characterized by progressive intracranial bone overgrowth, associated with a missense mutation (P.L441R) in the ZIP14 gene, which results in the mislocalization of the protein in osteoblasts, allowing high intracellular zinc accumulation, which causes excessive bone growth in the skull [347].

Additionally, ZIP8 is present on the cytoplasmic membrane as well as the intracellular vesicles of various cell types and is able to transport zinc from outside to inside. Its expression was found to be increased in the chondrocytes of OA patients and in OA mouse models, resulting in increased intracellular zinc concentrations, which upregulate the expression of zinc-dependent metalloprotein matrix-degrading enzymes, which degrade the extracellular matrix, leading to the onset of pathology [348].

ZIP13, on the other hand, is located at the vesicular and Golgi levels, where it acts by transporting zinc from the subcellular compartment to the cytosol. It has a wider distribution in the bone context, being found not only in osteoblasts but also in chondrocytes and fibroblasts [349]. Murine studies have shown that the deletion of ZIP13 negatively affects signaling transduction by TGFβ/BMPs, which via both canonical Smad-dependent pathways [350] and non-canonical Smad-independent signaling pathways [351] regulates *Runx2* transcription. Therefore, inadequate osteoblastic differentiation and thus impaired bone development were found [252]. A homozygous recessive mutation in the ZIP13 gene is known to cause a spondylocheiro dysplastic form of Ehlers–Danlos syndrome (SCD-EDS) in humans, which is an inherited connective tissue and bone disease [352].

Regarding the ZnT family within the bone context, the roles of ZnT5 and ZnT7 have been highlighted. ZnT5 is expressed in the Golgi, while ZnT7 is localized not only in the Golgi apparatus but also in vesicular compartments; both mediate the efflux of zinc from the cytosol to these compartments.

The role of the ZnT5 transporter in bone has not been fully elucidated, although in vivo studies showed that ZnT5 KO mice showed poor bone growth, osteopenia and heart failure. Mice deficient in this gene showed poor growth and a decrease in bone density due to an impairment of osteoblast maturation to osteocytes [353].

In vitro studies on MSC cells have highlighted that the overexpression of ZnT7 decreased the expression of the osteoblast *ALP*, and *Col-1*, as well as calcium deposition. In contrast, KO of ZnT7 promoted gene expression associated with osteoblast differentiation and matrix mineralization in vitro, such as the Wnt and ERK signaling pathways (Liu Y et al., 2013). Overexpression of ZnT7 protects MC3T3-E1 from H_2_O_2_-induced apoptosis. ZnT7, by mediating zinc entry, promotes cell survival through two distinct signaling pathways involving the activation of the protein kinase B (PI3K/Akt)-mediated survival pathway and activation of the ERK/MAPK pathway [354]. Overall, a few studies conducted on zinc transporters showed that their depletion correlates with the onset of several abnormalities. However, studies on their role in the pathogenesis of bone disease are rare. Further investigations are required to achieve a deeper understanding of their role at the cellular level during osteogenesis.

##### Pro-Osteogenic Action of Zinc

Several studies have reported data about the ability of exogenous zinc to upregulate the expression of bone early and late differentiation genes (Figure 7) in osteoblasts in a dose- and time-dependent manner.

In vitro studies carried out on osteoblastic cell line MC3T3-E1 showed that even a short period (24–72 h) of zinc exposure (within the range from 10^−6^ to 10^−4^ M) increases the expression of the main differentiation markers (RUNX2, OCC, Col-1α, OPG, regucalcin, ZIP1) [242]. Similarly, under prolonged (up to 10 days) cell exposure to Zn (with the range of 1 to 25 10^−6^ M zinc concentrations), Seo et al. observed that zinc can stimulate bone formation through the induction of proliferation and subsequent differentiation of osteoblasts, highlighting the increase in ALP activity (whose catalysis requires two zinc ions as cofactors) and collagen intra- and extracellular concentrations [355]. Accordingly, the MC3T3-E1 model has shown that zinc deficiency downregulates the expression of specific bone markers (Col-1, OPN, ALP, OCC) through a reduction and delay in the expression of the *Runx2* differentiation transcription factor [356], whose activity is regulated upstream by BMP2 [357,358]. Furthermore, this impacts on the decrease in matrix production and mineralization by osteoblasts, thus emphasizing the critical role that zinc plays in osteoblastogenesis [356].

In OB MC3T3-E1 lines, zinc depletion has been reported to suppress the expression of bone matrix genes and proteins, as also demonstrated by the decrease in ECM deposition (looking at Col-1, OCN and OCC as reporter genes) [359]. In addition, zinc deprivation has been reported to trigger a mitochondria-mediated apoptotic process in 75–90% of MC3T3-E1 cells (starting from a basal death rate of 7% under physiological conditions) [360]. A mirrored study conducted on human MSCs showed that a high level of zinc (>50 µM) increases *Runx2* expression levels, thus inducing the differentiation of stem cells to pre-osteoblasts [361].

Moreover, a positive effect of exogenous zinc in regulating osteoblastogenesis was also assessed on osteosarcoma Saos-2 cell lines, in which an increase in ALP activity was confirmed at different concentrations of zinc exposure (in the range of 1 to 10 µM, rather than 25–50 µM) [362].

During bone turnover, osteoblastic cells—in addition to being involved in signaling within the bone multicellular unit—literally move to occupy the site of resorption that was previously occupied by osteoclastic cells. Studies have therefore evaluated the role of zinc on osteoblast migration. High concentrations of Zn (≥200 µM) have been shown to act as a chemoattractive signal for MC3T3-E1 osteoblastic cells. Specifically, cell migration is directed to zinc-rich bone-resorbing sites [363]. Taken together, studies have shown that zinc has a pro-osteogenic effect on osteoblasts, regulating their differentiation and proliferation and inhibiting apoptosis in a dose- and time-dependent manner. Nevertheless, the concentration of zinc at which a positive effect on osteoblast activity can be expected in vitro occurs in a narrow dose range (1–50 µM) depending on the cell model used. Conflicting data exist even within the same cell model, presumably due to the form of zinc source (e.g., zinc acetate, zinc chloride), whether it is compounded with other proteins to facilitate cell endocytosis (presence of albumin or not) and the zinc exposure time in relation to the concentrations used. Regarding toxicity, zinc has the lowest toxicity for bone metabolism compared with other trace metals. Only the use of very high doses (600 and 900 µM) has made it possible to highlight the cytotoxic effects of zinc [364].

##### Exogenous Zinc as a Reinforcement for Endogenous Osteogenesis

Due to zinc’s involvement in regulating bone cell differentiation processes, many studies have been conducted in vivo to translate its use into a clinical context.

It is clear from such studies that a dietary deficiency of zinc disrupts the growth and development of bone in humans as well as in animal models, causing multiple bone abnormalities, including disturbances in bone formation, mineralization and hence the development of osteoporosis and osteoarthritis [365,366].

Zinc deficiency has been associated with an osteopenic bone phenotype, while low-dose intake or depletion has been associated with an increase in osteopenia, osteoporosis and fracture risk in men [367] as well as an increased risk of fracture and high bone loss in postmenopausal women [368,369]. Hence, it is not surprising that some clinical trials have picked zinc as an exogenous dietary supplement for inducing anabolic effects on bone.

Recent studies on mouse models have identified the action of zinc at the level of two fundamental pathways: (I) canonical Wnt signaling through β-catenin promotion of osteoblastogenesis [370] (see Section 3.1), and (II) the key RANKL/RANK pathway of bone turnover, which involves both osteoclast and osteoblast cells [371] (see Section 3.2).

In particular, mouse models have shown that a Zn-deficient diet reduces the number of OB precursors and of mature osteoblasts, thus reducing bone formation. Osteoblastogenesis reduction has been seen to be involved in the inhibition of Wnt/β-catenin signaling consequent to suppression of glycogen synthase kinase 3 beta (GSK3β) and activation of Akt [372], an important mitogenic signaling pathway that has a critical regulatory function in bone formation and remodeling [90].

It has been demonstrated that zinc has a suppressive effect on RANKL-induced osteoclastogenesis in mouse marrow cell culture. At the same time, it seems that zinc also enhances the OB expression of OPG, which further blocks the RANKL action [242]. Overall, the zinc action has subsequently been confirmed to inhibit bone resorption by concurrently acting on signaling that allows osteoclast differentiation, and by stimulating the apoptosis of mature osteoclasts, demonstrating its dual role in inhibiting osteoclast activation and maturation, promoting bone anabolism [242]. Hence, it has been suggested that zinc may exert protective properties against bone loss by suppressing osteoclastogenesis through the downregulation of the RANKL/RANK axis [371].

Further studies have evaluated the Zn effect on bone stability and on bone turnover processes at the metaphyseal–epiphyseal region in mouse models. The upregulation of remodeling markers and the concomitant decrease in resorption activities in relation to increasing exogenous zinc supplementation (2.5 to 30 μg/g) support a positive involvement of zinc in modulating the balance between bone formation and bone resorption [373]. Conversely, zinc deficiency is associated with negative skeletal outcomes. Studies on animals fed with a diet lacking or partially lacking in zinc have shown a decrease in bone development [374]. Later research has shown that feeding zinc-free mice also affects the decreased concentration and functionality of other trace elements that are essential for the functionality of other tissues, such as calcium and magnesium in the muscle and liver [375].

Taken together, studies have shown that zinc has a pro-osteogenic effect on osteoblasts, regulating their differentiation and proliferation and inhibiting apoptosis in a dose- and time-dependent manner.

In vivo studies carried out in animal models as well as trials conducted on humans have shown a general positive influence of zinc on the skeletal system in agreement with in vitro studies, further highlighting the pro-osteogenic activity of zinc.

### 5.2. Antioxidant Supplements Involved in Bone Metabolism

#### Effects of Phytochemicals on Bone Health

Compounds such as antioxidants influence intracellular redox homeostasis and can regulate bone formation and resorption by affecting redox-sensitive elements that are involved in the differentiation signaling pathway [376,377].

Many studies have reported the effect of various dietary antioxidant supplements in both osteoblastogenesis and osteoclastogenesis for the discovery of potential therapeutic agents.

Quercetin, a naturally occurring flavonoid found in onions and other vegetables, has beneficial effects on bone cells and tissues; its administration contrasts bone loss in ovariectomized mice [378] and in rat models of diabetic osteopenia [379]. In in vitro studies on MG-63 osteoblasts, quercetin was capable of increasing ALP activity [380].

Betulinic acid has been shown to stimulate the mineralization and differentiation of osteoblastic MC3T3 cells, probably through the activation of the BMP/Smad/RUNX2 and beta-catenin signal pathways [381]. The antioxidant apocynin, a natural compound structurally related to vanillin, exerts an accelerating effect on the differentiation of osteoblasts and suppresses the production of the bone-resorbing factors in MC3T3-E1 cells [382]. Other compounds such as the flavonolignan silibinin and N-acetylcysteine have been indicated as potential therapeutic agents by promoting bone formation and suppressing RANKL-induced osteoclastogenesis [171,383,384,385]. The antioxidant α-lipoic acid attenuates osteoclast differentiation by reducing NF-κB DNA binding and also suppresses bone resorption in vivo. Rotenone inhibits the osteoclastogenesis of primary precursor cells by regulating MAPK and transcription factor signaling pathways, and the in vivo efficacy of rotenone has been confirmed in animal models.

##### Resveratrol

Resveratrol (RSV; 3,40,5-trihydroxy-transstilbene) is a small polyphenol found in many plants, commonly used as a nutraceutical in the management of high cholesterol, cancer, heart disease and many other conditions [386]. Additionally, RSV has been shown to have multiple bioactivities including antioxidative, anti-inflammatory, estrogen-like and proliferative properties that can influence bone metabolism [387,388].

The literature suggests that RSV affects osteoclasts and osteoblasts either directly or indirectly by stimulating bone formation and decreasing bone resorption.

Boissy et al. showed that RSV dose-dependently stimulates the mRNA expression of the two osteoblastic markers osteocalcin and osteopontin in immortalized osteoblast-like human MSC-TERT cells [389]. Other studies have reported enhanced proliferation and differentiation of mouse osteoblastic MC3T3 cells [390,391,392], and the promotion of osteoblast differentiation from mesenchymal stem cells, acting on various signal transduction pathways [392,393].

In particular, RSV-treated osteoblastic MC3T3-E1 cells exhibited increased DNA synthesis and ALP activity, indicating a direct stimulation of osteoblast proliferation and differentiation by RSV [394]. The ability of the anti-estrogenic drug tamoxifen to antagonize these effects indicated that RSV stimulated osteoblastogenesis by acting as an estrogen agonist [390]. Supporting data suggested that RSV directly stimulates cell proliferation, osteoblastic differentiation and osteogenic gene expressions through mechanisms involving an ER-dependent pathway, and coupling to ERK1/2 activation in human MSCs [391].

RSV is also able to increase the expression of the key osteogenic transcription factors RUNX2 and osterix, decrease the expression of peroxisome proliferator-activated receptor gamma (PPAR-γ) and inhibit PPAR-γ by mediating the nuclear receptor corepressor, resulting in the promotion of the osteogenic differentiation of mesenchymal stem cells [382,391,394]. Additional studies have reported that RSV activates both the canonical Wnt signaling pathway and AMPK and reduces the formation of bone-resorbing osteoclasts by inhibiting NF-κB transcription activity, resulting in reduced RANKL-induced osteoclast differentiation [392,394].

Taken together, in vitro evidence indicates that RSV influences estrogen-dependent and independent signal transduction pathways which modulate the gene expression of transcription factors in bone cells [394] (Figure 8). Moreover, the ability of RSV to act on both osteoblasts and osteoclasts through multiple mechanisms suggests that RSV can prevent bone loss associated with different etiologies and pathologies.

In fact, in vitro findings and animal models suggest that RSV can be effectively beneficial in treatments of bone disease [395]. Thus, 12 weeks of treatment with RSV prevented ovariectomy-induced bone loss in rats [396,397], and bone loss following hind limb immobilization [398,399,400]. In addition, chronic RSV supplementation in mice prevented age-related deterioration in bone mineral density compared to those fed a standard diet [401]. Reduced inflammation and thus reduced bone resorption, in addition to increased bone formation, are suggested as potential explanations for these bone-protective effects. Further in vivo experiments by Xuhao and coworkers revealed that RSV treatment significantly improved bone quality and reduced the levels of serum ALP and osteocalcin in osteoporotic rats [402].

Data reported thus far suggest that RSV supplementation may be beneficial for bone health to prevent the age-related decline in functional integrity or as an experimental medicine in disorders of excessive bone destruction. However, the complex and convoluted intracellular mechanisms activated by RSV stimulation remain largely unknown.

Among the various molecular targets of resveratrol, two regulators of mitochondrial function, namely, silent information regulator of transcription1 (SIRT1), and Peroxisome proliferator-activated receptor-gamma coactivator (PGC)-1alpha, have been intensively studied [403,404,405]. SIRT1 is a class III nicotinamide adenine dinucleotide (NAD^+^)-dependent HDAC which also deacetylates non-histone cytoplasmic substrate proteins, such as p53 and NF-κB, to fine-tune normal cell epigenetics [406,407,408]. Through these activities, SIRT1 regulates important longevity-related processes including apoptosis, cell survival, DNA repair and energy expenditure. SIRT1 has been shown to be activated by resveratrol treatment and has also been proved to prevent aging-related diseases such as osteoporosis [409]. The effects of RSV on SIRT1 influence its interactions with RANKL and the bone-specific transcription factor RUNX2, in bone-derived cells and MSCs, respectively [392,410] (Figure 8).

Furthermore, it has been observed that administration of resveratrol ameliorates lipopolysaccharide (LPS)-inhibited osteoblast differentiation in MC3T3-E1 cells, which was accompanied by increased cellular SIRT1 and peroxisome PCG 1 [411].

Moreover, RSV promotes skeletal growth through an SIRT1–BMP2 longevity axis and protects osteoblasts by activating the PI3K/Akt/mTOR signaling pathway through enhancing mitophagy, via upregulation of SIRT1 expression in osteoporosis rats [402,412].

A recent work by Wang et al. substantiated this notion and provided further information on the role of resveratrol for the treatment and prevention of the damage that occurs due to postmenopausal osteoporosis [413]. In particular, the authors explored the regulatory effect of resveratrol on autophagy in osteoblasts and osteoclasts in a rat model of postmenopausal osteoporosis. They demonstrated that the inhibition of autophagy in osteoblasts and its activation in osteoclasts were reversed with resveratrol treatment, indicating the beneficial effects of RSV [413].

Collectively, evidence from animal models stresses the double actions of RSV on both osteoblasts and osteoclasts, supporting a therapeutic value of RSV supplementation on bone. To date, only a few clinical studies have evaluated the effects of resveratrol on BMD, but their results were lackluster, and none of the trials continued for more than 6 months [414,415]. Recently, a 24-month randomized, placebo-controlled trial revealed that regular supplementation with 75 mg of resveratrol twice daily improves BMD and reduces the bone resorption marker CTX-1 in postmenopausal women. Interestingly, the benefits of resveratrol on the spine and hip BMD appear to be amplified in women who regularly consume vitamin D and calcium supplements [416].

Further mechanistic-focused studies would improve our understanding as would experimental designs using comparable doses, timings of exposure and treatment durations. No toxicity has been reported for RSV intakes of up to 500 mg/d in animals and humans [387,417]. Due to its multiple bioactivities and low toxicity, RSV is a promising efficacious and safe therapeutic agent for osteoporosis. Meanwhile, evidence generated by animal studies will provide the necessary foundation for future clinical trials.

##### Coenzyme Q10

Coenzyme Q10 (CoQ10), also known as ubiquinone, is a lipid-soluble antioxidant which plays a role in the electron transport chain involved in the generation and regulation of cellular bioenergy [418,419]. In contrast to other lipophilic antioxidants, CoQ10 stems from endogenous synthesis and food intake. Rich sources of dietary coenzyme Q10 include mainly meat, poultry and fish.

CoQ10 demonstrates membrane-stabilizing activity and is a powerful antioxidant with free radical scavenging activity and cell-protective effects. As an antioxidant, CoQ10 is also capable of recycling and regenerating other antioxidants such as tocopherol and ascorbate [420]. The efficacy of CoQ10 supplementation for the treatment of human diseases has been widely studied, revealing the protective role of CoQ10 in heart failure, cancer, muscular dystrophy and nervous system disorders [421,422,423].

Several studies have reported that CoQ10 can dampen osteoclastogenesis and promote osteoblastogenesis, suggesting its potential therapeutic applications for the treatment of bone diseases. Moon and coworkers investigated both the promoting effect of CoQ10 on osteoblastogenesis in MC3T3-E1 cells and its inhibitory effects on RANKL-induced osteoclastogenesis in both bone marrow-derived monocytes (BMMs) and RAW 264.7 macrophages. They found that CoQ10 suppresses osteoclast differentiation by scavenging intracellular ROS [424], thus attenuating ROS-induced pathways for osteoclastogenesis signaling and NFATc1 gene expression.

Additionally, CoQ10 enhances bone regeneration at all differentiation stages through transcription factor activity, enhancing not only early osteoblastic biomarkers such as ALP and Col-1 but also late osteoblastic biomarkers such as BSP and matrix mineralization through transcription factors RUNX2 and OSX. CoQ10 also promoted matrix mineralization by enhancing bone nodule formation in a dose-dependent manner [424,425].

A more recent study by Zheng et al., based on in vitro and in vivo experiments, demonstrated that CoQ10 supplementation promotes the proliferation and differentiation of rat bone MSCs, in a dose-dependent manner, with an increased expression of osteoblastogenic markers, including RUNX-2, OCN and ALP. Moreover, CoQ10 effectively suppressed ovariectomy (OVX)-induced bone loss in rats, by reversing osteoporotic changes and maintaining the bone structure. The above-mentioned effects of CoQ10 may be mediated through activation of the phosphatase and tensin homolog (PTEN)/PI3K/Akt signaling pathway [426]. Based on these studies, CoQ10 may have therapeutic implications in treating osteoporosis and other bone diseases.

### 5.3. Exosomes in Bone Metabolism

The scientific community considers secretomes, in general, and exosomes, in particular, as promising disease diagnostic markers and drug delivery vehicles. Exosomes seem to support the regenerative and immunomodulatory abilities of MSCs during tissue repair.

#### 5.3.1. Exosome Vesicles

Exosomes are a class of extracellular small membrane-enclosed vesicular particles (secreted nanospheres with a diameter of 30–100 nm) found in almost all fluids and biological tissues, MSCs and bone cells [427,428], where they help the regulation of metabolism and intercellular communications in both physiological and pathological conditions. The number, size and content of exosomes can vary according to the cells of origin, the presence or absence of pathologies and the microenvironmental conditions. As with message-containing bottles, exosomes are vesicles (small enough to move freely in our body) capable of fusing with phospholipid bilayers to deliver messages to the target cell.

#### 5.3.2. Exosome Content

Exosomes contain various macromolecules from cytoplasmic synthesis, including proteins and coding and non-coding RNAs such as mRNAs, microRNAs (miRNAs) and long non-coding RNAs (ln RNAs) [402,429].

Although the exosomal protein composition varies according to the origin of cells and tissues, proteome profiling studies have shown that the diversity in proteins is rather limited. Exosomes do not contain nuclear, mitochondrial, endoplasmic reticulum and Golgi apparatus proteins, and the exosome proteins identified to date are found on the plasma membrane, in the cytosol or on membranes of endocytic compartments [430].

The protein cargo capacity of a single exosome, given certain assumptions of protein size and configuration, and packing parameters, can be about 20,000 macromolecules. The type of surface proteins, their carrying capacity and their stability make exosomes excellent extracellular messengers able to securely reach long-distance cells within our body and play an important role in physiological and pathological processes [431,432]. Mass spectrometry analysis has identified a total of 1536 proteins contained in osteoblast-derived exosomes; among those importantly involved in membrane trafficking and signaling pathways, we can mention transforming growth factor beta receptor (TGFβR) 3, LRP6, bone morphogenetic protein receptor type-1 (BMPR1) and Smad ubiquitylation regulatory factor-1 (SMURF1) [433].

It has been found that miRNAs play key roles in cell proliferation, differentiation, organ and tissue development and the regulation of bone homeostasis. They contribute to bone formation and resorption, bone remodeling and differentiation of bone cells. Importantly, the easy sampling and the long stability of exosome particles mean exosomes have great potential as biomarkers for various diseases, including osteoporosis.

#### 5.3.3. Exosome Biogenesis and Release

Although the processes through which specific bioactive molecules are packaged into exosomes are largely unknown, a specific mechanism generates exosomes when the inner membrane of the endosomes sprouts inward to form luminal vesicles, which then transform into multivesicular bodies (MVBs). At this point, when the MVBs do not undergo lysosomal degradation, they fuse with the cell membrane, releasing the exosomes’ cargo (Figure 9) [434,435].

Different from the membrane of microvesicles, the exosome membrane is similar to a plasmamembrane cell, as it is enriched with lipids such as cholesterol, sphingomyelin and ceramides [436]. Exosome loading depends not only on the cell type but also, more importantly, on the cell’s environmental stimulations: such as mechanical clues, cellular pH, biochemical stimuli and hypoxia [437,438,439].

#### 5.3.4. Role of Exosomes in Bone Remodeling and Molecular Mechanisms Involved

In the context of bone tissue, some compelling studies suggested that bone cells such as osteoblasts, osteoclasts and bone MSCs secrete exosomes, which not only serve for cell-to-cell communication within tissue, in several physiological processes, but are also important in several pathological conditions of skeletal disorders. As reported above, bone-derived exosomes contain a specific composition of molecules (such as proteins and nucleic acids) that vary dynamically according to cell types as well as pathological and physiological status.

Stromal MSCs play a fundamental role in osteoblastic differentiation not only as precursors but also as paracrine mediators through the secretion of regulatory exosomes which, endocytosed by osteoblasts, promote osteogenesis. Bone MSC-derived exosomes have been reported to bind and tether ECM proteins such as Col-1 and fibronectin to the bone surface and scaffolds [148] and upregulate *TGFβ1* expression, with BMP9 being a strong inducer of osteogenic differentiation [440].

As mentioned above, in general, non-coding RNAs, particularly miRNAs, are of most importance in the regulation of bone metabolism (Table 5). Specifically, miR-196a, miR-27a and miR-206, once transferred to OBs, induce the expression of key genes such as *Runx2* and *Alp*, promoting differentiation and osteogenesis [441]. The powerful regulatory effect of the long coding RNA MALAT1 (metastasis-associated lung carcinoma transcript 1) has also been demonstrated, which is found in large quantities in exosomes extracted from primary human MSCs. Specifically, MALAT1 directly targets miR-34c (a master transcription factor that dampens osteoblast proliferation and differentiation inhibitor). Among the miR-34c target RNAs, it is important to mention AT-rich sequence-binding protein2 (SATB2), activation transcription factor 4 (ATF4) and RUNX2, all of which are involved in the regulatory loop of OB differentiation, in turn enhancing bone formation [402]. The in vitro efficacy of exosomes in supporting osteogenesis has also been demonstrated in vivo in osteoporotic mice (i.e., oophorectomy-induced osteoporosis model) [438].

Mature osteoblasts are large producers of exosomes which are functionally active in the regulation of the osteogenesis process, and their cargo can modulate the timing of differentiation stages of their own cells of origin. Several miRNAs have been reported to be abundant in mineralized MC3T3-E1-derived exosomes, including miR-30d-5p, miR-133b-3p and miR140-3p, which are particularly dominant in bone tissue remodeling as they participate in Wnt, insulin, TGF and calcium signaling pathways [442]. It has been reported that at the terminal phase of MC3T3-E1 maturation, the osteoblast exosome cargo, containing miR196a mir-335-3p and miR-378b, strongly promotes differentiation on primary MSCs, targeting the Homebox C8 (HOXC8), DKK1 and CS pseudogene 3 (CSP3) genes, respectively [442,443,444]. Furthermore, this selective effect is also maintained in vivo, where they attenuate osteoporosis in ovariectomized mice [445].

MS identification of exosomes from OBs has revealed that proteins are predominantly involved in localizing structural proteins (such as protein phosphatase PP1C), intracellular signaling (e.g., RANKL and OPG) and histospecific enzymes such as TRAP [446,447]. Eukaryotic initiation factor2 (EIF2) is omnipresent, whereas there is sometimes some dysregulation of the exosomal expression of two NFkB-related genes, namely, a disintegrin and metalloprotease (ADAM) 17 and NFkB1 [448].

Moreover, exosomes represent an important means of osteoblast–osteoclast communication with both pro- and anti-osteoclastogenic effects. On the one hand, in mouse models, the exosomes released by osteoblasts are rich in RANKL (which promotes osteoclastic genesis and survival, see Figure 1b) [449]. On the other hand, OB exosomes can block the formation of osteoclasts through miR-125b which, when released into the matrix, is then captured by osteoclasts and inhibits PRDM (PR domain zinc finger) proteins which regulate transcription and microRNA genes. PRDM (PRDI-BF1 and RIZ homology domain-containing) protein family members are characterized by the presence of a PR domain and a variable number of Zn finger repeats. These may regulate the expression of proteins involved in extracellular matrix development and maintenance, including fibrillar collagens, such as Col4a1 and Col11a1, and molecules regulating cell migration and adhesion, including TGFβ2 [450]. PRDM proteins are also known as pro-osteoclastogenic transcription factors, which are targets of NFATc1, a crucial member of the transcription factor NFAT family [451]. A recent work described how osteoclasts also use exosomes to regulate bone remodeling, acting both on osteoblasts and on their own formation process [438]. In fact, a powerful regulatory mechanism has been demonstrated by miR-23a-5p, which is contained in large quantities in osteoclastic exosomes, which, in culture, is able to inhibit RUNX2, repressing osteoblast genesis (see Figure 1a and Section 2.2.1). Osteoclasts also release vesicles with numerous miRNAs and the RANK receptor which, on entering competition with its ligand (soluble or on the membranes of osteoblasts), reduces its bioavailability and, therefore, attenuates osteoclastogenesis which, in this way, is strictly controlled by negative feedback. The RANKL/RANK axis could also be exploited to convey molecules to osteoblasts (see Figure 2 and Section 3.2) [452].

Importantly, osteocytes, the master mechanosensor cells in regulating bone remodeling, release exosomes when stimulated biochemically and mechanically [441]. A mechanism of action has recently been identified in which exosomal miR-218 attenuates osteogenesis by blocking the synthesis of key proteins of the Wnt pathway in osteoblasts (such as sclerostin and transcription factor (TCF7)) (see Section 3.1).

#### 5.3.5. Possible Applications

From a therapeutic point of view, exosomes also have potential: cells could be the center of targeted therapies based on specific exosomes, transformed into carriers of active ingredients.

It has been discovered that MSC-derived exosomes could have a strong impact on cell therapies in regenerative medicine as these small subcellular structures can overcome many problems deriving from the use of live and expanded cells as therapeutic agents. Employing exosomes, all problems related to adverse phenomena such as the “protein corona” (i.e., a protein layer formation around artificial nanoparticles exposed to biological liquid) or other negative reactions related to the use of synthetic particles introduced into the human body could be overcome.

Cheap and effective strategies have been developed for the purification of exosomes from various biological fluids and the supernatant of cell cultures. The most common techniques used include centrifugation, chromatography, filtration, polymer-based precipitation and affinity chromatography [453,454,455,456]. Both analytical and preparative ultracentrifugation techniques are largely employed for the purification of exosomes derived from bone tissue [302]. Each purification technique exploits specific characteristics of exosomes, and the size, density, shape and enriched presence of surface proteins are parameters that can be used to facilitate and improve their isolation.

Therefore, exosome treatments in regenerative medicine could potentially be safer, more effective and cheaper than therapies based solely on MSC administration.

Increasingly more studies have investigated a possible therapeutic use in bone disease and in post-fracture regeneration, mainly by exploiting exosomes of mesenchymal cells. These are, for example, able to stimulate the growth of chondrocytes in models of osteogenesis imperfecta and to increase tissue repair after a fracture (through the involvement of miR-21, miR-4532 and miR-125b-5p) [457]. Furthermore, MSC exosomes obtained from the human umbilical cord or adipose tissue have shown great efficacy in animal studies in promoting regeneration or osteogenesis in models of osteoporosis, osteonecrosis or fracture [458,459]. Hence, exosomes could represent an alternative to stem cell transplantation for bone regeneration. Although most of the data available thus far come from animal studies, it has been shown that human exosomes (purified from human umbilical cord plasma) can also promote osteogenesis and inhibit osteoclast genesis in mouse models of osteoporosis [460].

One of the main problems to be solved for the potential use of exosomes as therapeutic agents is their short half-life in the circulation. In fact, they tend to accumulate mainly in the liver and lungs. Additionally, in this case, however, various resolution strategies are already being studied, such as the addition of specific aptamers for MSCs, which have produced excellent results in vivo, reducing bone mass loss and improving regeneration in fracture patterns [461].

Exosomes could also represent adjuvants to other therapeutic strategies, enhancing their effectiveness. For example, in bone repair, titanium nanotubes modified with BMM-derived exosomes treated with BMP2 can significantly increase angiogenesis and osteogenesis and could, therefore, represent new biomaterials [445]. A similar improvement in bone repair has been obtained with scaffolds of tricalcium phosphate (β-TCP) modified with human induced pluripotent stem cell-derived mesenchymal stem cell exosomes, following activation of the calcium signaling of PI3K/Akt in MSCs [462]. It should be added that exosomes can be the target themselves. It has been shown that blocking the release of RANKL-rich osteoblastic exosomes (via imipramine) protects mice from ovariectomy-induced bone loss [463]. In other words, in physiological conditions, the regulation of bone remodeling is completely in accordance with that of parent cells. Thus, in principle, an aberrant endosomal sorting could possibly be specific for a given pathology. Therefore, the analysis of the bone-derived exosomal content of specific pathologies is of clinical interest.

## 6. Conclusions and Future Directions

Bench-to-bedside strategies for bone augmentation should be conceived according to further elucidation of the biomechanics and molecular mechanisms involved in bone repair. The evolving discipline of mechanomics focuses on physical forces and their impact on the cellular and pericellular molecular mechanisms. A major challenge in the mechanobiology interdisciplinary field is to mechanobiologically understand the mechanisms by which mechanical signals are transduced into a cascade of biochemical events [464], and to understand how these molecular events contribute to development, physiology and disease. In this regard, extensive research has reported that aberrations in mechanotransduction pathways result in disease-like effects (e.g., deregulated mechanoresponsive signaling in osteoarthritis and osteosarcoma). Unfortunately, only a few signaling pathways have actually been described to be involved in the development of bone diseases [465].

The evolution of the field of bone mechanobiology, from tissue-level studies to investigations at cellular levels, has improved our fundamental knowledge. These studies have revealed two pathways heavily involved in translating mechanical influences into a biochemical cellular anabolic response (i.e., Wnt and RANK/RANKL/OPG signaling).

Substantial evidence indicates that the Wnt signaling pathway participates in the transduction of mechanical signals at the cell surface and ultimately leads to the regulation of bone metabolism. However, the anabolic mechanisms that are triggered by physical forces in human bone cells at the cellular level remain unclear.

The soluble context is another important determinant factor, besides the specific mechanical signal employed as an anabolic agent. Indeed, hormones influence the effect of the cell response to mechanical stimulation, while soluble Wnt signaling ligands, such as SOST, act as coupling factors between biomechanical and biochemical osteoanabolic responses (Figure 4). Hence, for the development of novel anabolic strategies for bone therapy, it would be useful to exploit multiple environmental beneficial agents.

Nutrition is one of several important modifiable factors for optimal bone health and prevention of osteoporosis. The correlation between the intake and/or serum levels of several vitamins and bone health in humans has been extensively reported with heterogeneous findings, showing positive, negative or negligible effects. Vitamin D is essential for calcium absorption and bone mineralization which is positively associated with BMD [182]. The role of vitamin A remains controversial; excessive, as well as insufficient, levels of retinol intake may be associated with compromised bone health [192,199]. Despite limited evidence, deficiency in vitamin K seems to be related to bone loss, decreased bone strength and an increased risk of fracture [277,286,299]. In general, the relationship between vitamins and bone is complex and seems to be affected by genetic factors, gender, menopausal status, hormonal therapy, smoking and calcium intake [187]. Moreover, it has been suggested that dietary and nutritional patterns may be more important than the intake or level of individual vitamins in bone health, thus explaining some of the paradoxical results related to individual nutrients.

Among dietary supplements, several antioxidants have been shown to be effective in bone formation and resorption by affecting redox-sensitive elements that are involved in the differentiation signaling pathway [376,377]. Notably, in vitro findings and animal models indicate that RSV and CoQ10 can dampen osteoclastogenesis and promote osteoblastogenesis, suggesting their potential therapeutic applications for the treatment of bone diseases, also by virtue of their low toxicity [394,401,402,425,426]. However, the complex and convoluted intracellular mechanisms activated by their stimulation remain largely unknown.

Among the trace metals, zinc certainly deserves further investigation in order to define the optimum concentration and time exposure to be employed for promoting bone anabolism. Investigations at cellular levels have reported that zinc may have an anabolic effect through the binding of zinc finger motifs of osteoblast transcription factors (RUNX2 and OSX), regulating their differentiation and proliferation and inhibiting apoptosis in a dose- and time-dependent manner [25]. The pro-osteogenic activity of zinc has been proved through in vivo studies as well as in trials, and a general positive influence of zinc on the skeletal system has been demonstrated. In this respect, it has been suggested that trace ion supplements and vitamins regulate gene transcription and differentiative signaling pathways via interaction with a multitude of other factors [249,250]. These findings highlight the importance of adequate nutrition in preserving bone mass and reducing the risk of osteoporosis and fractures.

To discover novel strong inducers of osteogenic differentiation, researchers have focused on the secretome screening of bone-derived exosomes. However, in order to develop exosomes as drug delivery vehicles, their direct clinical application problems related to their short half-life in the circulation should be tackled.

Overall, as cells and tissues concurrently sense all modifications of environmental physical-chemical properties, reacting to adapt their physiological response, multifactorial experimental approaches should be considered to search for new therapies. However, the combinatorial effect of multiple microenvironmental cues with mechanical stimuli is not trivial; thus, it still remains poorly investigated. Indeed, further research on the interplay and synergism between mechanotransduction processes and conventional soluble biochemical osteomodulators may uncover additional soluble factors which may have therapeutic potential in preventing and treating bone disease. Nonetheless, elucidation of the molecular cascades and crosstalk following mechanical stimulation under physio-pathological conditions would guarantee further steps forwards successful treatments. An applicative strategy could be considering the effects of ossification agents on bone cells in the context of interaction with other hormones, growth factors, bone morphogenetic factors and other signaling molecules, evaluating the potential therapeutic applications of nutritional supplements and vitamins for the treatment of bone disease.

Taking all these studies together, it appears that research should focus on promising soluble ossification agents that may crosstalk with the Wnt signaling pathway (via regulation of RANKL/OPG), creating a permissive environment which is able to boost mechanical stimulation effects. Furthermore, clarification of the molecular signaling of molecular pathology will facilitate the development of reliable prognostic/diagnostic tools as well as novel treatment strategies in bone diseases.

## Figures and Tables

**Figure 1 cells-10-02383-f001:**
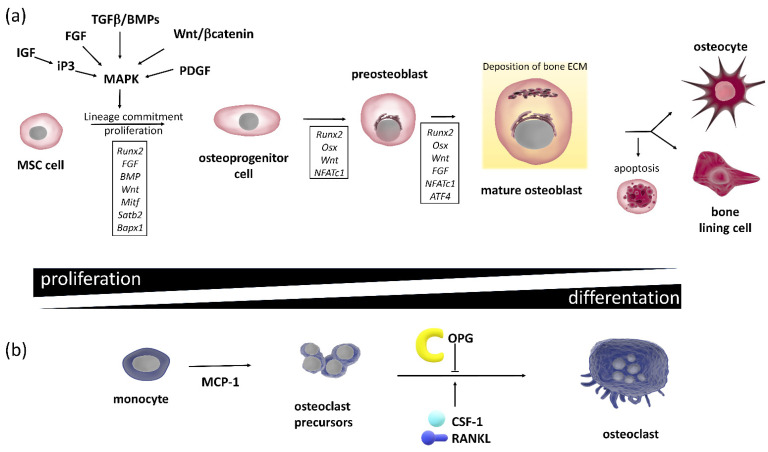
The principal signaling networks and transcription factors regulating bone cell differentiation. (**a**) Osteoblastogenesis and osteogenesis. In boxes, transcriptional factors, which characterize each stage of osteogenic differentiation, are shown. The MSC population actively proliferates at the initial stages of osteogenesis. As MSCs commit to osteoblasts, their proliferation rate decreases while they start expressing osteogenic genes. Following mineralization, mature osteoblasts undergo apoptosis, revert back to a bone lining phenotype or become embedded in the mineralized matrix and differentiate into osteocytes. Lines with an arrowhead indicate a positive action, and lines with a bar indicate inhibition. (**b**) Osteoblast cytokines involved in osteoclastogenesis: osteoblasts produce chemokine MCP-1 (monocyte chemoattractant protein-1). In addition, osteoblasts express the master of osteoclastogenesis cytokines, i.e., CSF-1 (light blue sphere), RANKL (represented in dark blue) and OPG (yellow semicircle). Monocytes (differentiated from HSCs) evolve to osteoclast precursors and finally to active OC forms which are stimulated by RANKL. Together with the canonical Wnt signaling, the RANK/RANKL OPG signaling pathways control osteoclasts in response to the actual extracellular stimuli.

**Figure 2 cells-10-02383-f002:**
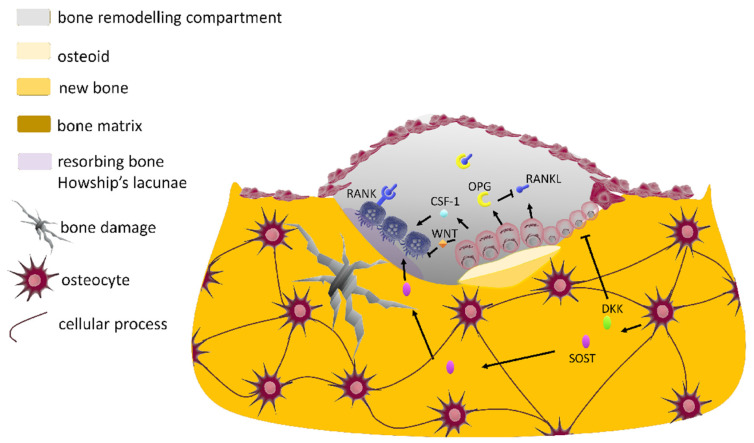
Molecular crosstalk in bone molecular unit (BMU): Paracrine actions of osteoblast-, osteocyte- and osteoclast-derived factors within the bone remodeling compartment. Osteoblasts respond to external signals generated by mechanically activated osteocytes or direct endocrine signals, recruit osteoclast precursors to the remodeling site, by expressing CSF, RANKL (represented in dark blue) and WNT (orange diamond), and inhibit osteoclast activity through OPG (yellow semicircle), a decoy receptor of RANKL (pictured in dark blue). Osteocyte-derived SOST (magenta oval) inhibits OB differentiation and stimulates osteoclastogenesis. The osteocyte expression levels of Wnt inhibitors (SOST and DKK (green oval)) temporally control the cycle of bone remodeling. Lines with an arrowhead indicate a positive action, and lines with a bar indicate inhibition.

**Figure 3 cells-10-02383-f003:**
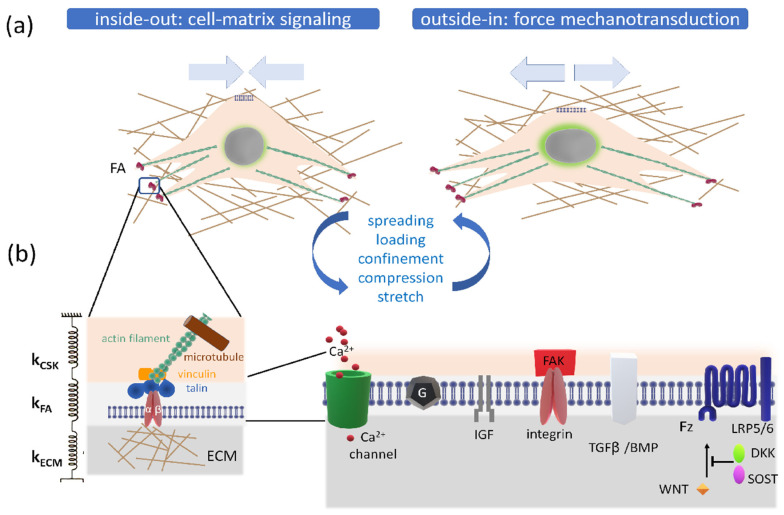
(**a**) Representation of inside-out and outside-in mechanotransduction signals. Focal adhesions (FAs) serve as crucial sites for transferring forces in both directions. Integrins are coupled to the cytoskeleton via molecules such as vinculin, talin and α-actin. (**b**) Protein network clusters across the extracellular matrix, transmembrane proteins and cytoskeleton regions of a spread cell. On the right side, three nonlinear spring series conceptualize the mechanical linkage between the cytoskeleton, focal adhesion complex and extracellular matrix, with respective nonlinear spring constants: k_CSK_, k_FA_, k_ECM_. On the left side, a zoom of the membrane portion is represented. Mechanical signals perceived by membrane-bound receptors such as stretch-activated Ca^2+^ channels, integrins, G proteins, IGF and TGFβ and/or BMP receptors are stimulated by mechanical forces and converted into a proper biological response (Table 2). The ECM and intracellular pathways are biochemically coupled by mechanotransduction pathways: mechanical resistance to intrinsic forces regulates the stability of focal adhesion complex that contains focal adhesion kinase (FAK), which phosphorylates and activates mechanoresponsive signaling elements. Line with an arrowhead indicates a positive action, and line with a bar indicates inhibition.

**Figure 4 cells-10-02383-f004:**
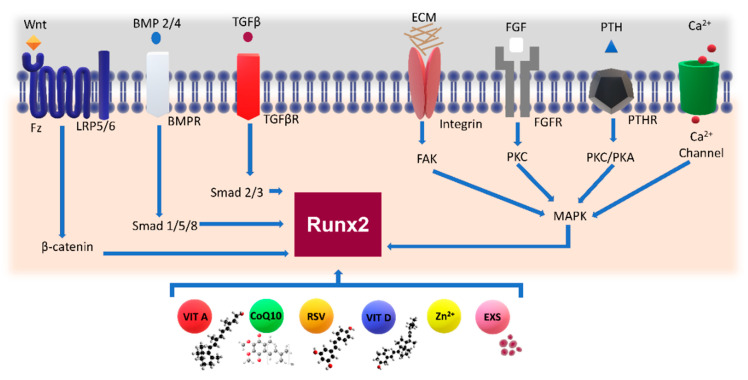
The regulation of Runx2 by mechanical and soluble ossification agents. Mechanical signals perceived by membrane-bound receptors such as lipoprotein-related protein (LRP) 5/6 and the frizzled (Fz) co-receptors, TGFβ and/or BMP receptors, integrins, FGF and G proteins (as an example, the PTH receptor is indicated) and stretch-activated Ca^2+^ channels regulate Rnux2 activity. Major regulatory pathways are represented. Biochemical agents such as vitamin A, coenzyme Q10, resveratrol (RSV), vitamin D, zinc ion and exosomes (EXS) can affect Runx2 and thus osteoblast differentiation.

**Figure 5 cells-10-02383-f005:**
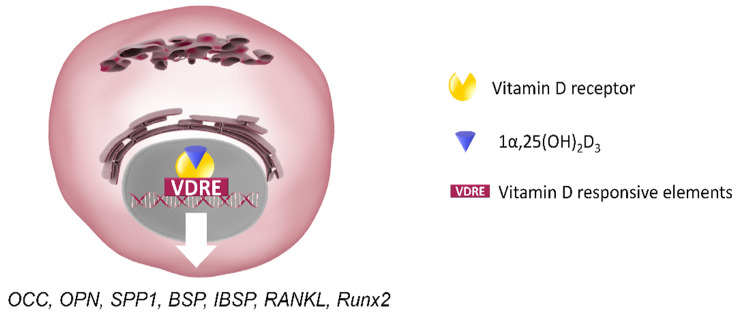
Scheme of direct actions of 1,25(OH)2D3/VDR on mature osteoblasts. 1,25(OH)2D3 acts via the VDR to regulate the osteoblast-related genes containing VDRE binding motifs. 1,25(OH)2D3 can both positively and negatively regulate expression of osteoblast phenotypic markers as a function of the proliferative and differentiated states of osteoblasts and the duration and concentration of exposure. SPP1, Secreted Phosphoprotein 1; BSP, bone sialoprotein; IBSP, integrin binding sialoprotein.

**Figure 6 cells-10-02383-f006:**
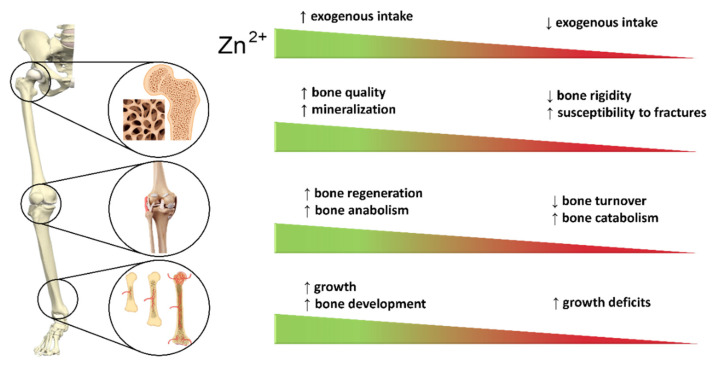
Effect of zinc intake (at high or low concentrations/doses) on the skeletal system.

**Figure 7 cells-10-02383-f007:**
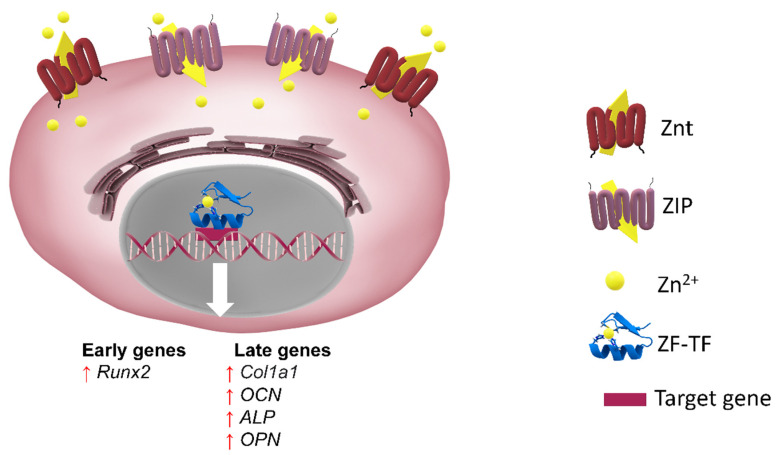
Representation of the zinc transport process across membranes, mediated by specific importers and exporters, to Zrt- and Irt-like protein (ZIP) and the zinc transporter (ZnT). At the cytoplasmic level, zinc is important for the functionality of zinc finger transcription factors (ZF-TFs) that regulate the transcription of early and late genes in osteoblastic differentiation. Refer to the text for a detailed description.

**Figure 8 cells-10-02383-f008:**
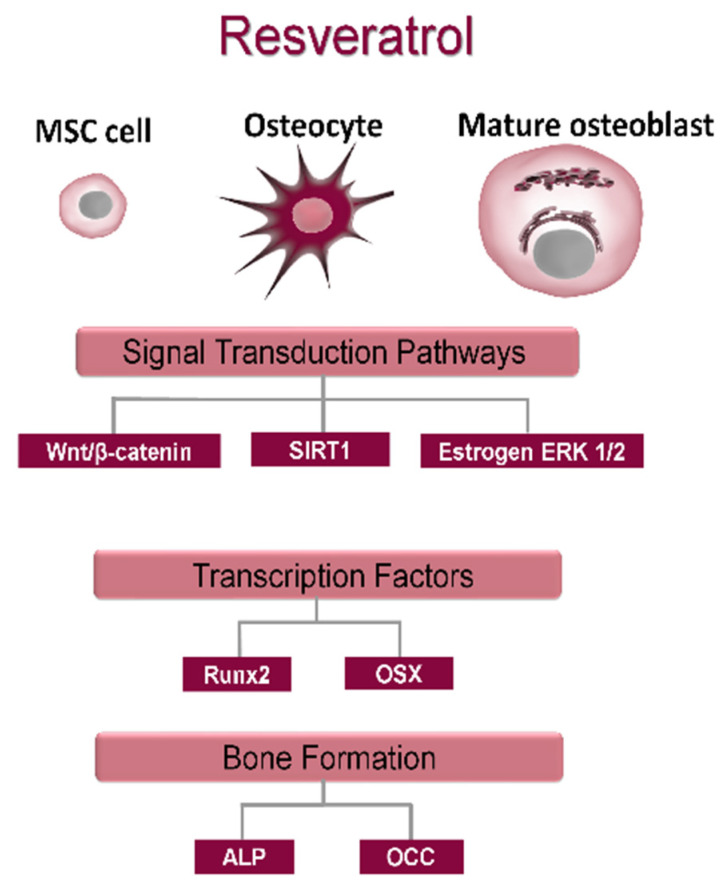
Osteogenic effects of resveratrol in vitro. RSV influences estrogen-dependent and independent signal transduction pathways which modulate the gene expression of transcription factors Runx2 and osterix (OSX), regulating osteoblast differentiation and mineralization.

**Figure 9 cells-10-02383-f009:**
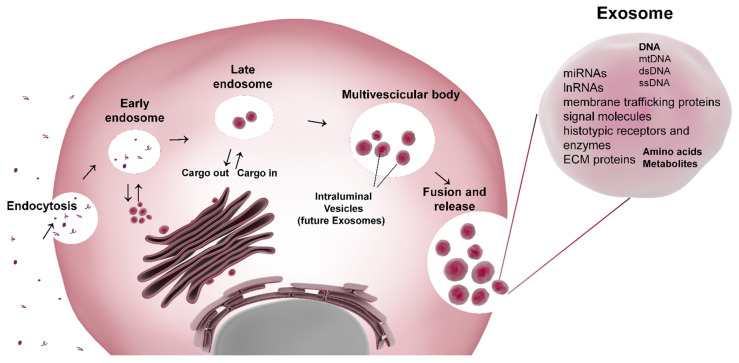
Exosome biogenesis. Through endocytosis and plasma membrane invagination, fluid and extracellular constituents such as proteins, lipids, metabolites, small molecules and ions can enter cells. This process leads to the formation of early endosomes or the possible fusion of the gem with endosomes performed by the constituents of the endoplasmic reticulum and of the Golgi network. Early endosomes give rise to late endosomes. Late endosomes evolve to multivesicular bodies (MVBs) with a defined collection of intraluminal vesicles (future exosomes). MVBs can also fuse directly with lysosomes for degradation (not shown). MVBs that do not follow this pathway can be transported to the plasma membrane. Exocytosis follows and leads to the release of exosomes with a lipid bilayer orientation similar to that of the plasma membrane. Exosomes can contain different types of cell surface proteins, intracellular proteins, RNA, DNA, amino acids and metabolites.

**Table 1 cells-10-02383-t001:** Functions and cell signaling of specialized bone cells involved in the bone remodeling process.

Cell Type	Description	Major Functions	Key Signaling and Pathways
Osteoblasts	differentiate from MSCs but may also derive from bone lining cells [31]; may form a low columnar “epithelioid layer” at sites of bone deposition; are polarized cuboidal cells containing plenty of rough endoplasmic reticulum and large Golgi apparatus [32]; are responsible for bone calcification; once mature, cannot divide and have three possible fates: they can become a bone lining cell or an osteocyte or undergo apoptosis (Figure 1a) [33]	osteoid formation: secretion of type I collagen-rich bone matrix and regulation of matrix mineralization [34]	the RUNX2 transcription factor starts osteoblastogenesis [23]; OSX, a zinc finger transcription factor, regulates transition from osteoprogenitors to pre-osteoblasts; The canonical Wnt signaling pathway promotes OB differentiation, and it is antagonized by the secreted proteins SOST and members of the DKK family synthesized by osteocytes (Figure 1b) [24,35,36,37]; Hedgehog signaling, NOTCH, FGF and BMP [38] promote OB differentiation (Figure 1a)
Osteocytes	most abundant cells in bone, >90% of all adult bone cells [33]; derive from mature OBs that, once the osteoid (unmineralized matrix) is mineralized, terminally differentiate into osteocytes end up residing in small lacunae inside the calcified bone matrix; stellate cells with long dendritic processes that ramify in canaliculae; throughout the mineralized bone matrix, interconnection of osteocytes (Figure 1b) is mediated by GAP junctions, connecting osteocytes to bone lining cells and bone marrow cells, in a complex intercellular network [38]	mechanosensor cells that transduce bone loading signals to orchestrate the action of BMU [39,40]; are also involved in mineral homeostasis [41]	major source of RANKL required for osteoclastogenesis during bone remodeling [42,43]; secrete SOST and DKK-1, the negative regulators of Wnt signaling that limit osteoblastic bone formation (Figure 1a); secretion of SOST and DKK-1 is inhibited by mechanical loading, and thus an increased loading corresponds to a local apposition of bone mineralization (Figure 1b) [44]
Osteoclasts	multinucleated cells formed by fusion of precursors (derived from HSCs) that share precursors with macrophages; podosomes facilitate adhesion to the bone surface and formation of a sealing zone, providing an isolated acidic resorption bay within which OCs can dissolve calcium salts into soluble forms and digest the bone matrix [45]	bone minerals are dissolved though acidification, and bone matrix is broken down by secretion of lysosomal enzymes that proteolyze organic ECM [46]	differentiation is initiated by M-CSF factor and promoted by RANKL; upon the binding to its cognate receptor RANK on precursor cells [45], osteoclastogenesis is negatively regulated by osteoblast-derived decoy receptor OPG which binds RANKL to inhibit its binding to RANK (Figure 1b) [47]
Osteoprogenitor cells	flat squamous cells located in the periosteum (external surface) and endosteum (internal surface) undifferentiated cells can divide to replace themselves can become osteoblasts	constant replenishment of these osteoblastic lineage cells	Ras-MAPK pathway regulates EPK signaling to form the skeletal structure, regulating differentiation of osteoprogenitor cells without changing proliferation [48]; signals transduced by TGFβ superfamily members control the formation of tissue differentiation; further, BMPs activate Smad 1 and 5 as extracellular signals through their effects on cell proliferation, differentiation and migration [49]
MSCs	once activated by active TGFβ, they migrate to bone-resorptive sites; can differentiate into osteoblastic lineage		all osteoblast progenitor cells present SOX9 transcription factor [38]
Pre-osteoblasts	heterogeneous population of cells, including those transitioning from MSC cells to mature osteoblasts which express RUNX2	are a key player in the osteogenic process	mechanistic target of rapamycin (mTOR) integrates both intracellular and extracellular signals to regulate cell growth and cell differentiation [50]
Bone lining cells	post-mitotic, long-lived flat osteoblast lineage cells lining the bone surface	can be a source of OBs in response to anabolic stimuli [31]	Wnt signaling [51]

**Table 2 cells-10-02383-t002:** Mechanotransduction signaling. When a bone is mechanically loaded, cells detect the physical deformation, converting the perceived mechanical strain signal into a biological output (i.e., a cellular response). Although the precise biochemical pathways have yet to be fully unraveled, different response pathways have been reported to mediate the adaptative response to mechanical loading and unloading in bone.

Signal Mechanotransduction Mediators	Effects	References
Mitogen-activated protein kinase (MAPK) signaling pathway	Increases RUNX2, osterix, eNOS osteopontin, osteocalcin and CoX2 and MMP13 expression; RANKL downregulation; increases osteoblast commitment; ATP-dependent activation of calcium channels; integrin activation	[82,91,106,107]
PI3K/Akt signaling	Important mitogenic signaling which provokes rapid increase in intracellular calcium levels; activation of IP3, ATP and NO; release of PGE2	[98,102,108,109,110,111]
G protein-mediated signaling	Activation of heterotrimeric GTPases via G protein coupling receptor rises intracellular calcium; cAMP and cGMP activation of rhoA GTPases	[96,112]
Wnt/beta-catenin pathway	Increases bone density; the amount of beta-catenin decreases, thus increasing its cytoplasmic concentration, possibly potentiating beta-catenin nuclear translocation; downregulation of sclerostin, thus increasing OB activity	[88,113,114,115,116]
Prostaglandins and prostacyclin (eicosanoid-derived phospholipids)	Their exogenous administration stimulates bone formation and increases the sensitivity of bone to external loads (PGE2); their release occurs concurrently with NO; PGE2 increases GAP junction communication and the formation of focal adhesions	[103,117,118]
Nitric oxide	Induces activity of NO synthase	[119,120]
Stromal cell-derived factor 1 (SDF-1)	Induces differentiation and recruitment of mesenchymal cells; influences cell adhesions and migration	[121,122]
Nucleotide signaling	Release of ATP into extracellular space; calcium mobilization; upregulation of RUNX2	[123]
Estrogens	Activation of TGF1 receptor; COX2 gene is induced; ERα a downregulates sclerostin expression, whereas ERβ decreases the osteogenic response to loading	[124,125,126,127]

**Table 3 cells-10-02383-t003:** Mechanical parameters employed for the description of physical stresses in mechanobiology.

Parameter	Description	Symbol	Unit
Loading pressure	the mechanical stress is a measure of load per unit of area	P	Pa (N/m^2^)
Strain	the ratio of change in length to the original length, when a given body is subjected to some external force (expressed in percentage: change in length/the original length)	ε µε	% % × 10^−6^
Frequency	number of applied cycles per second or per minute	n w	Hz (1/s) cycles/min
Strain rate	temporal change in strain magnitude within each strain cycle	µε/s	1/s
Strain distribution	spatial change in strain magnitude across a given volume	Δµε/d	
Strain volume	expresses the total number of daily loading cycles	cpd	cycles/day

**Table 4 cells-10-02383-t004:** Effects of trace metals on bone metabolism.

Trace Nutrients	Sources	Bone Effects
Boron	It is present mostly in soil and water, meaning the dietary sources are plant-based such as vegetables, fruits and nuts [303]	↑ Mineralization [304] ↑ Regeneration of bone [305,306]
Copper	The best dietary sources are cereals, whole grain products, seeds, nuts and chocolate, as well as shellfish and animal offal [307]	↑ Matrix stability and strength [308] ↑ Bone differentiation [309] ↑ Bone remodeling [310]
Iron	Foods containing the highest amounts of iron are red meat, especially offal, shellfish, pulses, fruits and especially nuts [307]	Maintains bone homeostasis [311]
Fluorine	It is present in soil and water; consequently, fruits and vegetables may contain traces of it [312]	↑ Bone mass and density [313] ↑ Osteoblastogenesis [314,315]
Selenium	The main source of selenium is a proper diet, meaning the right selection of animal and plant products [316]	↑ Protection against oxidative stress [317] ↑ Bone mass [318]
Chromium	Good sources are meat and whole grain cereals, some fruits and some vegetables [319]	↓ Mineralization [320] ↑ Oxidative stress [321]
Cobalt	The main sources of Co in the diet are fish, green leafy vegetables and cereals [322]	↓ Bone modeling [323] ↑ Oxidative stress [321]
Cadmium	The environment and smoking are the two main sources of Cd exposure in humans, specifically from contaminated food or drinking water [324]	↑ Fracture risk [325] ↓ Bone formation [324] ↑ Bone resorption [324]

**Table 5 cells-10-02383-t005:** Role of exosomes in bone remodeling. Cell of origin and target cell are specified for each exosomal molecule. BMSC, bone mesenchymal stem cell; OB, osteoblast; OC, osteoclast; BMM, bone marrow monocyte.

Cell of Origin	Exosomal Cargo	Target Cell	Biological Effect
BMSC	miR-196a, miR-27a, miR-206	OB	↑ osteogenesis
BMSC	MALAT1	OB	↑ osteogenesis
BMSC	miR-122-5p	OB	↑ osteogenesis
BMSC	not specified	OB	↑ osteogenesis
OB	miR-667-3p, miR-6769b-5p, miR-7044-5p, miR-7668-3p, miR-874-3p, OPG	BMSC	↑ osteogenesis
OB	ECM proteins (tenascin C, fibronectin, collagen, TRIP1)	ECM; BMSC	↑ osteogenesis
OB	RANKL, TRAP	BMM	↑ OC genesis
OB	miR-125b	OC	↓ OC genesis
OB	MMP2	Endothelium	↑ angiogenesis
OC	miR-214-3p	OB	↓ osteogenesis
OC	miR-23a-5p	OB	↓ osteogenesis
OC	RANK receptor	OC	↑ osteogenesis
Osteocyte	SOST, RANKL, OPG	OB	↑ osteogenesis
Osteocyte	miR-218	OB	↓ osteogenesis

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
