# Peer review of "Effects of Extracellular Osteoanabolic Agents on the Endogenous Response of Osteoblastic Cells"

_cells, 2021, doi:10.3390/cells10092383_

Round 1

Reviewer 1 Report

This manuscript presents an overview of the extracellular osteoanabolic agents on the endogenous response of osteoblastic cells. The mechanical stimulation and some of the ossification coactivators including vitamin A, vitamin D, vitamin K, CoQ10, Zinc and exosome are discussed in this article. This is an interesting review which would help to better understand the role of extracellular stimuli in skeletal metabolism, but there is room for improvement in several areas. The manuscript is not well-written, with lots of non-standard expressions, wrong formatting, missing punctuations, and logical inconsistencies. The authors need to revise and polish the whole manuscript carefully. The following concerns include some more detailed comments and suggestions for modifications that the authors should consider.

  1. This manuscript is not easy to read, since the organization and logic in some part appear confusing in parts. For example, in the section 2.2 (Page 17, Line 670-672), the authors use the words “on the other hand” and “therefore”, while I cannot understand how to link them to the above. Likewise, can OPG bind to RANK and inhibit the secretion of RANKL? If so, please add the references here. And how to come to the conclusion “the availability and interaction of RANKL/RANK/OPG determines the efficiency of osteoclastogenesis” according to the above? Besides, there is no need to describe “RANKL is produced by several bone cell types” twice (Page 17, Line 661 and 669), it makes no sense here. There are also some other places which make me confused, so it would be better if the authors could carefully go through the whole manuscript and improve the organization and expression logic.
  2. A proper introduction is necessary for the reader to better understand the background, which should not be missed in the manuscript. Please add it.
  3. This review is exceedingly long with significant redundancy. For example, there are too many descriptions about the osteoblast physiology and function, and the common introductions of vitamins and exosome. There is no need to provide so much unnecessary basic knowledge, and it would be better to focus instead on the latest research in the field.
  4. The manuscript is full of short paragraphs. For example, there are 14 paragraphs with 2-3 sentences per paragraph in the section 2.1.4. It is suggested that the authors merge short paragraphs of similar themes so that the reader can quickly locate the information they need.
  5. Page 17, Line 675: I did not find the description about the association between Wnt signalling and RANK/RANKL/OPG signalling pathways in Table 2. Please have a check on it and cite correctly.
  6. There are several mechanosensitive ion channels which play important roles in bone cell mechanotransduction, such as transient receptor potential vanilloid (TRPV) channel and the piezo family of proteins. These essentially need to be discussed in the review.
  7. The ‘conclusions’ section is too long and contains a large amount of research background and repetition of already described text elements.

Minor Comments:

  1. Page 2, Line 71-79: These sentences represent the instructions for authors, which definitely should not be placed in the manuscript.
  2. Page 2, Line 91 and Page 36, Line 1587: The sentences should end with a period.
  3. Page 10, Line 339 and Page 26, Line 1105: I think the “Holle et al 2001” and “Stafford et 1105 al., 2005” here are the wrong cite format. Please check again the references carefully for proper formatting.
  4. Page 16, Line 608 and 643: The words "VDR-mediated" are in the different format with bold and smaller font.
  5. Page 14, Line 516: Why is the term "osteocytes" underlined?
  6. Page 3, Line 125-126: “Growth Factor ” and “Platelet-Derived Growth Factor ” are capitalized in the first letter, while the “insulin-like growth factors” and “fibroblast growth factor ” are not. Please keep the expression in the manuscript consistent.
  7. Page 3, Line 153: The “andli ght” should be “and light”.
  8. Page 4, Line 113-114: The “(OB)s ” and “(OC)s” should be “(OBs) ” and “(OCs)”.
  9. The terms “remodeling” and “remodelling” are both correct, but they need to be consistent within the manuscript depending on British English or American English, respectively. Please check them carefully.

Author Response

Re:Reviewer 1

This is an interesting review which would help to better understand the role of extracellular stimuli in skeletal metabolism, but there is room for improvement in several areas. The manuscript is not well-written, with lots of non-standard expressions, wrong formatting, missing punctuations, and logical inconsistencies. The authors need to revise and polish the whole manuscript carefully. The following concerns include some more detailed comments and suggestions for modifications that the authors should consider.

Replay: We thank the reviewer for stating that the work is a significant contribution to the field and for her/his several suggestions which have helped authors to improve the manuscript.

 This manuscript is not easy to read, since the organization and logic in some part appear confusing in parts. For example, in the section 2.2 (Page 17, Line 670-672), the authors use the words “on the other hand” and “therefore”, while I cannot understand how to link them to the above. Likewise, can OPG bind to RANK and inhibit the secretion of RANKL? If so, please add the references here. And how to come to the conclusion “the availability and interaction of RANKL/RANK/OPG determines the efficiency of osteoclastogenesis” according to the above? Besides, there is no need to describe “RANKL is produced by several bone cell types” twice (Page 17, Line 661 and 669), it makes no sense here. There are also some other places which make me confused, so it would be better if the authors could carefully go through the whole manuscript and improve the organization and expression logic.

Replay:. The manuscript has been carefully revised in agreement with the Reviewer’s suggestions in order to make it more fluent and clear, where necessary. The text has been reorganized and two new sections (i.e., section 0  and section 3.4)  and an additional figure( figure 4 of the revised version) have been introduced. Where possible the text has been shortened.

  1. A proper introduction is necessary for the reader to better understand the background, which should not be missed in the manuscript. Please add it.

Replay: Following the reviewer advise, in the revised version the text has been reorganized adding an additional section entitled: 0. Introduction

This review is exceedingly long with significant redundancy. For example, there are too many descriptions about the osteoblast physiology and function, and the common introductions of vitamins and exosome. There is no need to provide so much unnecessary basic knowledge, and it would be better to focus instead on the latest research in the field.

Replay: The detailed descriptions about cell biochemistry concepts were reported in the manuscript as the authors believed these could be helpful for the wide interdisciplinary fields of mechanobiolology, particularly for biophysicists and engineers. We structured the manuscript in several sections and subsections so that the reader can quickly locate the information of interest. However, we see that the paper ended up being extensively long. Therefore, following the reviewer suggestions we have shortened the text by removing (or reducing) the basic knowledge information throughout the text.

  1. The manuscript is full of short paragraphs. For example, there are 14 paragraphs with 2-3 sentences per paragraph in the section 2.1.4. It is suggested that the authors merge short paragraphs of similar themes so that the reader can quickly locate the information they need.

Replay:  As suggested, the short paragraphs have been merged throughout the text.

Page 17, Line 675: I did not find the description about the association between Wnt signalling and RANK/RANKL/OPG signalling pathways in Table 2. Please have a check on it and cite correctly.

Replay: The referee is right since the association is not reported in table 2; the text has been corrected accordingly.

  1. There are several mechanosensitive ion channels which play important roles in bone cell mechanotransduction, such as transient receptor potential vanilloid (TRPV) channel and the piezo family of proteins. These essentially need to be discussed in the review.

Replay: A paragraph on the mechano-sensitive transient receptor potential vanilloid (TRPV) channel and the piezo family of proteins has been added in 1.3.3 section.

  1. The ‘conclusions’ section is too long and contains a large amount of research background and repetition of already described text elements.

Replay: The ‘Conclusions and future directions’ section has been shortened and partially rephrased following Reviewer’s suggestion. For easier reading, some information has been moved in a new section (namely: 3.4 Cell response to anabolic mechanical treatments).

 Minor Comments:

  1. Page 2, Line 71-79: These sentences represent the instructions for authors, which definitely should not be placed in the manuscript.

Replay: We erased those sentences refused from the format file.

  1. Page 2, Line 91 and Page 36, Line 1587: The sentences should end with a period.

Replay: We added the missed fullstop.

  1. Page 10, Line 339 and Page 26, Line 1105: I think the “Holle et al 2001” and “Stafford et 1105 al., 2005” here are the wrong cite format. (nel PDF è corretto!  Please check again the references carefully for proper formatting.

Replay: We corrected the citations.

  1. Page 16, Line 608 and 643: The words "VDR-mediated" are in the different format with bold and smaller font.

Replay: We corrected the font.

  1. Page 14, Line 516: Why is the term "osteocytes" underlined?

Replay: We corrected the font.

  1. Page 3, Line 125-126: “Growth Factor ” and “Platelet-Derived Growth Factor ” are capitalized in the first letter, while the “insulin-like growth factors” and “fibroblast growth factor ” are not. Please keep the expression in the manuscript consistent.

Replay: Done.

  1. Page 3, Line 153: The “andli ght” should be “and light”.

Replay: We corrected the typo.

  1. Page 4, Line 113-114: The “(OB)s ” and “(OC)s” should be “(OBs) ” and “(OCs)”.

Replay: We corrected as suggested.

  1. The terms “remodeling” and “remodelling” are both correct, but they need to be consistent within the manuscript depending on British English or American English, respectively. Please check them carefully.

Replay: We checked and corrected as suggested.

Reviewer 2 Report

This review by Alloisio et al. should be very interesting for other scientists working the field of bone biology and bone regeneration. However, I have found that this manuscript is in fact quite extensive with certain sections harboring long descriptions of known processes such as bone remodeling, the characteristics of the different cells found in bone tissue, the biochemistry of Vitamin D, etc. In fact, both the extension and the fact that the authors dedicate substantial parts of the manuscript to aspects that are covered in more generalist texts fits the description of a book chapter more than that of a review in a more specific scientific field.  The extensive and more introductory parts of the text could be easily reduced by introducing a suitable reference without this having any impact in the core of the work. I have specified some of them but the authors should look at reducing the size of most of the sections. This would substantially benefit their manuscript. On the contrary, I consider that the section dedicated to biomechanics is highly interesting as this is something not normally covered in more general reviews on bone biology.

Section 1.1. I have found the initial description about bone composition and cell types extremely extensive. All data found in this section can be found at general histology or biochemistry book. This section should be dramatically reduced. Please limit this section to the key information needed to build up following sections.

My previous comment about section 1.1 also applies to sections 1.2.1 and 1.2.2. The description of Bone histology and the process of bone remodeling should be reduced to a minimum. The extensive description provided by the authors in this section is not fundamental to understand subsequent sections.

In section 2, same as previously, the introduction about osteogenic signaling pathways is too extensive and detailed for a specialized scientific publication. Lines from 527 to 535 would be a good example of this. The author should try summarizing this and citing relevant bibliography were this is addressed instead of describing again processes that are already well stablished and widely known in the scientific field of bone regeneration.

Although the contents of section 4. are very interesting for researchers working in the bone field they might constitute an independent revision on its own. The authors have put a great deal of effort on this revision, but it seems to this reviewer that it might be too much to take for a single manuscript. The part corresponding to “Ossification coactivators” opens yet another extensive block of information that might as well be treated independently. Besides, speficifically regarding Section 4.1.2, some parts of this section might be redundant with section 2.1.4. Also, overall, rather than mixing the effect of the different “coactivators” on the different bone cells it would help if the effect of those coactivators on the different bone cells is structured in subsections.

Also in relation to this section, another important note is that the authors include Exosome vesicles inside this “ossification coactivator” part. Although this might be controversial, and due to the pleiotropic effect of these molecules a single miRNAs carried by the exosomes could on trigger a complete osteogenesis program, thus, treating the role of exosomes in this section might not be correct. Also, similarly to other sections, this can be importantly reduced. Exosomes main characteristics and biogenesis amongst other points covered in this section can be found elsewhere. There is no need to extensively address these points in this work.

As explained by the reviewers through the different sections, Runx2 not only has a key role in osteogenesis, but also is the protein where many of the signaling pathways involved in this process seem to converge. Perhaps a Figure reflecting this or a section specifically dealing with this point would be advisable.

There are several typos throughout the text. Although this reviwer understands that it is extremely difficult to find all the typos in such an extensive text, the authors should do their best at sending a better version of text from the spelling point of view. Also look out for double spaces.

Some specific comments.

Line 3. Eliminate “so”

Lines 71 to 79. There seems to be some text referring to instructions for introduction writting (?) that have been accidentally copied into the text. Please, remove these lines.

Line 80.  Bone remodeling is a process intrinsic to the tissue that occurs constantly (as actually mentioned by the authors in line 114) not only to accommodate changing mechanical stress or to repair fractures. Please, rephrase.

Line 112. Remaining instead remaining (typo)

Line 118: Please write: “ the first step in bone formation…”

Lines 143 to 145: Why is this sentence written in italics?

Regarding Figure 1A and text in section 1.2.2. While it is true that Runx2 expression is key to the progression of osteogenic differentiation, a sustained expression of this protein into later stages of this process has in fact a negative effect on the overall differentiation (Bruderer et al, Eur Cell Mater 2014). This should be at least mentioned in the text and indicated in the correspondent figure.

Description of MSCs in Table 1 is slightly poor. Also, What is the difference between osteoprogenitor cells and MSCs from the authors’ point of view? The authors use different terms that might be confusing to refer to mesenchymal stem cells. They are referred as mesenchymal progenitor cells at some point. This might be confusing for the reader.

Lines 288 to 298. This paragraph is somehow confusing. Do the authors mean that Runx2 activates BMP2 production? As far as this reviewer knows, the process is just the opposite. Please check.

Lines from 384 to 387. This sentence is very difficult to follow. Some punctuation signs might be misplaced. Please, rephrase

Line 549: “Osteoblast bone formation”?

Line 553: Give a brief explanation of this sentence. Reading the interpretation is that at some point osteoblasts must choose between differentiation and apoptosis as the only two ways of action. Is this the idea the authors want to transmit?

Section 2.1.4. The introductory part of the section (lines 582 to 601) is again too extensive and covers very general aspects of the biochemistry of Vitamin D.

Line 643. Part of the text is in written in bold.

Lines 673 to 675. This stamen seems to leave out other important routes implicated in osteoclast bone formation.

Line 719. Something is missing (verb?)

Lines 872 to 879. The authors mentioned the different effects of micromolar concentrations of ATRA on the C3H10T1/2 cell line and on the MC3T3 cell line. These differences might be justified by the different developmental state so those cells lines. Thus, these data might not be in disagreement. Eliminate “on the contrary” in line 878 to reflect this.

Line 968 seems to be redundant with previous statements.

Line 980 to 982. Do the authors refer to an overall reduction of  all gene expression?

Line 993. This is quite puzzling. Could the authors elaborate?

Project number is not indicated.

Author Response

This review by Alloisio et al. should be very interesting for other scientists working the field of bone biology and bone regeneration. However, I have found that this manuscript is in fact quite extensive with certain sections harboring long descriptions of known processes such as bone remodelling, the characteristics of the different cells found in bone tissue, the biochemistry of Vitamin D, etc. In fact, both the extension and the fact that the authors dedicate substantial parts of the manuscript to aspects that are covered in more generalist texts fits the description of a book chapter more than that of a review in a more specific scientific field.  The extensive and more introductory parts of the text could be easily reduced by introducing a suitable reference without this having any impact in the core of the work. I have specified some of them but the authors should look at reducing the size of most of the sections. This would substantially benefit their manuscript. On the contrary, I consider that the section dedicated to biomechanics is highly interesting as this is something not normally covered in more general reviews on bone biology.

 Section 1.1. I have found the initial description about bone composition and cell types extremely extensive. All data found in this section can be found at general histology or biochemistry book. This section should be dramatically reduced. Please limit this section to the key information needed to build up following sections.

 My previous comment about section 1.1 also applies to sections 1.2.1 and 1.2.2. The description of Bone histology and the process of bone remodelling should be reduced to a minimum. The extensive description provided by the authors in this section is not fundamental to understand subsequent sections.

 Replay:. The manuscript has been carefully revised in agreement with the Reviewer’s suggestions in order to make it more fluent and clear, where necessary. The text has been reorganized and shortened and two new sections (i.e., section 0 and section 3.4 ) have been introduced.

 In section 2, same as previously, the introduction about osteogenic signaling pathways is too extensive and detailed for a specialized scientific publication. Lines from 527 to 535 would be a good example of this. The author should try summarizing this and citing relevant bibliography were this is addressed instead of describing again processes that are already well stablished and widely known in the scientific field of bone regeneration.

Replay: The detailed descriptions about cell biochemistry concepts were reported in the manuscript since the authors believe these could be helpful for the wide interdisciplinary fields of mechano-biology, (particularly for biophysicists and engineers). We structured the manuscript in several sections and subsections so that the reader can quickly locate the information of interest. However, we see that the paper ended up being extensively long. Therefore, following the reviewer suggestions we reduced the basic knowledge information maintaining the key concepts needed to understand the following sections. Throughout the text corrections have been done accordingly.

The part corresponding to “Ossification coactivators” opens yet another extensive block of information that might as well be treated independently. Besides, speficifically regarding Section 4.1.2 (vitamin D), some parts of this section might be redundant with section 2.1.4 (Regulation of differentiative signaling pathways by vitamin D).

Replay: Following the reviewer advise, section 4.1.2 and 2.1.4 have been shortened.

Also, overall, rather than mixing the effect of the different “coactivators” on the different bone cells it would help if the effect of those coactivators on the different bone cells is structured in subsections.

Replay: Although the effect of the single coactivator is reported for different bone cell types, the discussion mainly focusses on osteoblastic cells (as also indicated by the title of the manuscript). Therefore, the authors would prefer to maintain this section structured as it is, as the suggested modification would be out of the scope of this paper.

Also in relation to this section, another important note is that the authors include Exosome vesicles inside this “ossification coactivator” part. Although this might be controversial, and due to the pleiotropic effect of these molecules a single miRNAs carried by the exosomes could on trigger a complete osteogenesis program, thus, treating the role of exosomes in this section might not be correct. Also, similarly to other sections, this can be importantly reduced.

Replay: Exosomes were considered in this review because these have a strong impact on cell therapies since they display a distinctive drug delivery capacity. In principle, they could possibly become excellent ossification activators. Therefore, they have a great potential use as therapeutic agents, as they could be transformed into carriers of active ingredients and employed in regenerative medicine.

Exosomes main characteristics and biogenesis amongst other points covered in this section can be found elsewhere. There is no need to extensively address these points in this work.

Replay: The 4.3 section has been modified according to the suggestions.

 As explained by the reviewers through the different sections, Runx2 not only has a key role in osteogenesis, but also is the protein where many of the signaling pathways involved in this process seem to converge. Perhaps a Figure reflecting this or a section specifically dealing with this point would be advisable.

Replay: Following the Reviewer’s advice we insert an additional figure. At the beginning of section 4 a new figure (Figure 4 in the revised version) illustrates that the different extracellular osteonabolic stimuli converge affecting runx2 activity.

 There are several typos throughout the text. Although this reviwer understands that it is extremely difficult to find all the typos in such an extensive text, the authors should do their best at sending a better version of text from the spelling point of view. Also look out for double spaces.

Replay: At the end of the revision the text has been carefully inspected.

  Some specific comments.

 Line 3. Eliminate “so”

Replay: Done.

 Lines 71 to 79. There seems to be some text referring to instructions for introduction writting (?) that have been accidentally copied into the text. Please, remove these lines.

Replay: Done.

 Line 80.  Bone remodeling is a process intrinsic to the tissue that occurs constantly (as actually mentioned by the authors in line 114) not only to accommodate changing mechanical stress or to repair fractures. Please, rephrase.

Replay: The sentence has been rephrased.

 Line 112. Remaining instead remaining (typo)

Replay: the typo has been corrected.

 Line 118: Please write: “ the first step in bone formation…”

Replay: Done.

 Lines 143 to 145: Why is this sentence written in italics?

Replay: The typo has been corrected.

 Regarding Figure 1A and text in section 1.2.2. While it is true that Runx2 expression is key to the progression of osteogenic differentiation, a sustained expression of this protein into later stages of this process has in fact a negative effect on the overall differentiation (Bruderer et al, Eur Cell Mater 2014). This should be at least mentioned in the text and indicated in the correspondent figure.

Replay: The suggested paper has been cited.

 Description of MSCs in Table 1 is slightly poor.

Replay: The authors believe that an exhaustive description of MSC is beyond the scope of this manuscript.

Also, What is the difference between osteoprogenitor cells and MSCs from the authors’ point of view?

The authors use different terms that might be confusing to refer to mesenchymal stem cells. They are referred as mesenchymal progenitor cells at some point. This might be confusing for the reader.

Replay: Done.

 Lines 288 to 298. This paragraph is somehow confusing. Do the authors mean that Runx2 activates BMP2 production? As far as this reviewer knows, the process is just the opposite. Please check.

Replay:  The authors agree with the referee, the sentences have been reworded. 

 Lines from 384 to 387. This sentence is very difficult to follow. Some punctuation signs might be misplaced. Please, rephrase

Replay: The punctuation signs have been corrected.

 Line 549: “Osteoblast bone formation”?

Replay: The typo has been corrected.

 Line 553: Give a brief explanation of this sentence. Reading the interpretation is that at some point osteoblasts must choose between differentiation and apoptosis as the only two ways of action. Is this the idea the authors want to transmit?

Replay: the sentence has been rephrased.

 Section 2.1.4. The introductory part of the section (lines 582 to 601) is again too extensive and covers very general aspects of the biochemistry of Vitamin D.

Replay: Following the reviewer’s suggestions, we have drastically reduced the introductory part of the section on vitamin D, only keeping the quotations, which might be useful for the reader.

 Line 643. Part of the text is in written in bold.

Replay: The typo has been corrected.

 Lines 673 to 675. This stamen seems to leave out other important routes implicated in osteoclast bone formation.

Replay: The sentence has been rephrased.

 Line 719. Something is missing (verb?)

Replay: This subsection describes two mechanical characteristics; the authors do not believe that the title needs a verb. In the new title we added the word “characteristic” which might help the reader.

 Lines 872 to 879. The authors mentioned the different effects of micromolar concentrations of ATRA on the C3H10T1/2 cell line and on the MC3T3 cell line. These differences might be justified by the different developmental state so those cells lines. Thus, these data might not be in disagreement. Eliminate “on the contrary” in line 878 to reflect this.

Replay: Done.

 Line 968 seems to be redundant with previous statements.

Replay: We removed it as suggested.

 Line 980 to 982. Do the authors refer to an overall reduction of  all gene expression?

Replay: The sentence has been reworded and the specific genes have been reported.  

 Line 993. This is quite puzzling. Could the authors elaborate?

Replay: The sentence has been erased since it was redundant .

 Project number is not indicated.

Replay:  Project numerber ,Beyond Borders 19 E84I20000620005, has been inserted.

Reviewer 3

The review provides a good overview on the topic. The review is well written and complete. 

I have no specific criticisms. 

Minor. line 339, please include de references in the reference list and indicate it as number. 

Replay: We thank the reviewer who has appreciated the great deal of effort employed by authors to review a such wide interdisciplinary fields of extracellular osteoanabolic stimulators.

Reviewer 3 Report

The review provides a good overview on the topic. The review is well written and complete. 

I have no specific criticisms. 

Minor. line 339, please include de references in the reference list and indicate it as number. 

Author Response

Reviewer 3

The review provides a good overview on the topic. The review is well written and complete. 

I have no specific criticisms. 

Minor. line 339, please include de references in the reference list and indicate it as number. 

Replay: We thank the reviewer who has appreciated the great deal of effort employed by authors to review a such wide interdisciplinary fields of extracellular osteoanabolic stimulators.

Round 2

Reviewer 1 Report

The quality of the revised manuscript has improved significantly. However, there are still some concerns that need to be addressed.

  1. In the first two sections, the revised manuscript in part still includes too detailed descriptions which are rather unnecessary for the readers to understand the main content of this review. For example, in the section 1.2, the general descriptions of osteoblasts (e.g. the lifespan of OBs) and bone biochemical markers are well known for every researcher working on bone tissues. For researchers working in other fields, these descriptions are not helpful to understand the core work of this review. Therefore, it is sufficient to summarize the main conclusions into several sentences.
  2. The ‘conclusions’ section remains too long.
  3. There are still lots of non-standard expressions and misspellings (examples are listed below), so this manuscript needs to be further edited to make it easier to read.

Minor Comments:

  1. Page 2, Line 63; Page 3, Line 74, Page 18, Line 681: There are two different abbreviations for “Nuclear factor of activated T cells” in the “Abbreviations” section. The NFATc1 should be the abbreviation for nuclear factor of activated T cells, cytoplasmic 1.
  2. Page 4, Line 147: The “the” should be “The”.
  3. Figure 1, Page 6, Line 227: The “monocite” should be “monocyte”.
  4. Table 1: “Responsible for bone calcification” should be “are responsible for bone calcification”.
  5. Table 2: The "mechano transduction" should be “mechanotransduction”.
  6. Page 29, Line 1227: The “works” should be “work”.
  7. Table 4: The “Is present mostly in soil” should be “It is present mostly in soil”.
  8. Page 3, Line 1662-1663: The “non coding” should be “non-coding”.
  9. Page 39, Line 1663; Figure 9.: The standard abbreviation of long non-coding RNAs should be “lncRNAs”.

Author Response

  1. In the first two sections, the revised manuscript in part still includes too detailed descriptions which are rather unnecessary for the readers to understand the main content of this review. For example, in the section 1.2, the general descriptions of osteoblasts (e.g. the lifespan of OBs) and bone biochemical markers are well known for every researcher working on bone tissues. For researchers working in other fields, these descriptions are not helpful to understand the core work of this review. Therefore, it is sufficient to summarize the main conclusions into several sentences.

Re: As requested we further shortened the 1.2 section. The authors believe that further reduction of the text would affect the comprehension of the manuscript for those readers who do not belong to the classical cell biology. The review regards an extremely wide interdisciplinary field, so to facilitate the reading it is necessary to introduce the manuscript defining the basic of both the biochemistry and the biomechanics of skeletal metabolism. The text has been structured in several sections and subsections, a reader expert in a field can quickly skip the known portions. Thus, only the introductory description, which gives the biochemical background for the biomechanical response, has been retained. The biochemical marker section has been reduced so to mention those which are extensively discussed in the following sections.

  1. The ‘conclusions’ section remains too long.

Re: A few paragraphs have been moved to section 4. We deem not to further reduce the text; otherwise important points would be missed.

  1. There are still lots of non-standard expressions and misspellings (examples are listed below), so this manuscript needs to be further edited to make it easier to read.

Re: All of non-standard expressions have been addressed. All minor comments have been addressed.

Reviewer 2 Report

The reviewer has no further comments.

Author Response

No  further comments are requested.